# MathCAMPS: Fine-grained Synthesis of Mathematical Problems From Human Curricula

## Abstract

Mathematical problem solving is an important skill for Large Language Models (LLMs), both as an important capability and a proxy for a range of reasoning abilities. Existing benchmarks probe a diverse set of skills, but they yield aggregate accuracy metrics, obscuring specific abilities or weaknesses. Furthermore, they are difficult to extend with new problems, risking data contamination over time. To address these challenges, we propose MathCAMPS: a method to synthesize high-quality mathematical problems at scale, grounded on 44 fine-grained "standards" from the Mathematics Common Core (CC) Standard for K-8 grades. We encode each standard in a formal grammar, allowing us to sample diverse symbolic problems and their answers. We then use LLMs to realize the symbolic problems into word problems. We propose a cycle-consistency method for validating problem faithfulness. Finally, we derive *follow-up questions* from symbolic structures and convert them into follow-up word problems—a novel task of mathematical dialogue that probes for robustness in understanding. Experiments on 29 LLMs show surprising failures even in the strongest models (in particular when asked simple follow-up questions). Moreover, we evaluate training checkpoints of Pythia 12B on MathCAMPS, allowing us to analyze when particular mathematical skills develop during its training. Our framework enables the community to reproduce and extend our pipeline for a fraction of the typical cost of building new high-quality datasets.

## 1 Introduction

As Large Language Models (LLMs) become increasingly capable, mathematical reasoning problems have emerged as a key benchmark for evaluating their abilities. Mathematical reasoning is a critical subproblem of many important tasks, such as scientific question answering and quantitative data analysis, making it a prerequisite for a range of downstream applications. Moreover, mathematical reasoning tests a broad spectrum of reasoning skills, serving as a valuable proxy for assessing reasoning capabilities more generally. Consequently, several benchmarks, notably GSM8K (Cobbe et al., 2021) and MATH (Hendrycks et al., 2021), became popular measures of the progress of LLMs, with each new generation of models demonstrating rapid advancements.

However, the classical approach to benchmarking in Machine Learning, which involves evaluating models on a fixed set of human-created problems, faces new fundamental challenges in the era of LLMs. First, these models are trained on massive public datasets that may unintentionally include the very benchmarks used for evaluation, raising concerns about data contamination (Zhang et al., 2024; Bubeck et al., 2023; Balloccu et al., 2024). This problem is exacerbated by the lack of access to the training data of most state-of-the-art LLMs, such as GPT-4 (Achiam et al., 2023), Claude (Anthropic, 2024), and even open-weight models, such as LLaMA (Touvron et al., 2023). Evaluating LLMs on novel problems could mitigate the data contamination concerns. But creating new mathematical problems is challenging. Crafting new high-quality problems requires expertise and is expensive; sourcing problems from public sources does not address the question of whether LLMs might have been trained on those problems.

Moreover, while existing benchmarks serve to track overall progress in the field, they do not inform us about what mathematical abilities current language models do and do not have. A single aggregate

---

*Equal contribution.

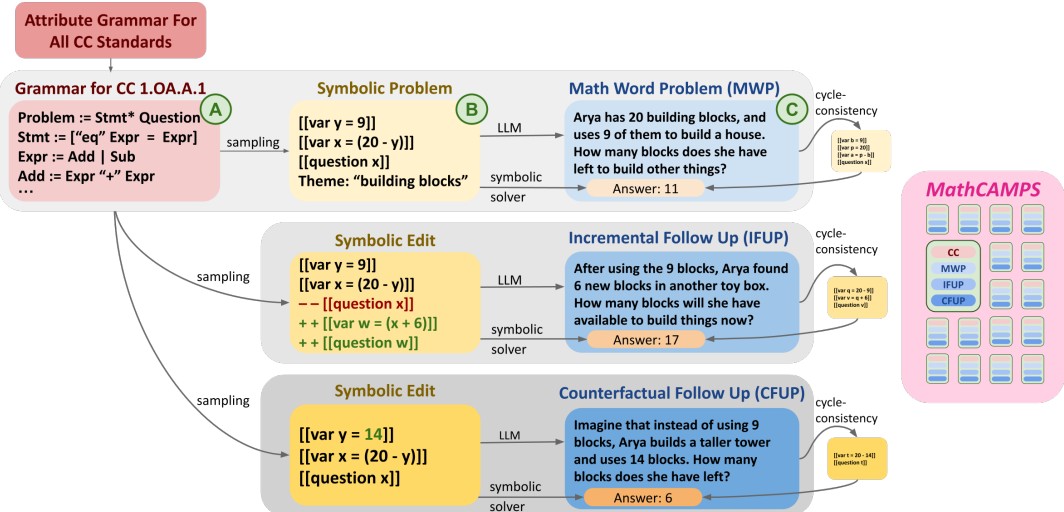

Figure 1: Overview of the MathCAMPS generation pipeline. We start from a grammar (**A**) that represents problems tied to a Common Core Standard - a specific mathematical ability drawn from a human curriculum. We sample problems in a symbolic form (**B**), and use a language model to realize it in natural language (**C**), applying a cycle-consistency where we back-translate the problem into symbolic form and ensure the answer remains the same, validating truthfulness. We also synthesize incremental and counterfactual follow-up problems

accuracy — in a topic as diverse as mathematics — does not provide insights into specific capabilities or challenges for current language models, and how those have been changing over time. For instance, GPT-4 (Achiam et al., 2023) improved by 35% on GSM8K when compared to GPT-3 (Brown et al., 2020); yet, it is still challenging to understand which improved capabilities might have accounted for this improvement (e.g., arithmetic with larger numbers, proficiency with fractions or decimals, or understanding of longer problems). Such an analysis would help shed light on open questions about language model learning, and how it relates to (or diverges from) human learning.

To address these challenges, we propose the Mathematics Common Core Assessment of Problem Solving — *MathCAMPS* — a framework for synthesizing high-quality mathematical word problems at scale. Our approach is grounded in the Mathematics Common Core (CC) Standards from Kindergarten through 8th grade. The CC standardizes a mathematics curriculum adopted by thousands of schools, describing specific abilities that students should learn by each grade. By constructing MathCAMPS in direct relation to the CC, our benchmark enables a series of rich analyses of mathematical proficiency in language models, allowing direct parallels to abilities that human students are also evaluated on. We encode the skills described in the CC (namely the standards) in a grammar that allows us to sample an arbitrary number of diverse problems targeting that skills (e.g., word problems involving addition of decimals, or solving systems of equations with fractions), represented symbolically.

Our pipeline uses a symbolic solver (SymPy) to obtain answers to the symbolic problems, and employs an LLM to realize those into word problems. We introduce a cycle-consistency method to validate whether a word problem faithfully represents the original symbolic problem. Prompting the LLM to back-translate the word problem into a symbolic structure and comparing the new answer to the original enables us to eliminate most unfaithful generation errors and maintain high quality.

Furthermore, building on our symbolic representation of problem structures, we introduce a novel task of "mathematical dialogue". In this task, once the LLM answers a problem correctly, we ask a follow-up question to further probe understanding. We introduce two types of follow-up problems: *counterfactual*, where we modify an aspect of the original problem and request an updated answer, and *incremental*, where we provide additional information and ask for a new answer. These questions require simultaneously understanding the original problem and the LLM's own solution — an additional challenge that several models struggle with.

Using our framework, we synthesize problems for each of 44 CC standards, resulting in a dataset of 4,900 initial problems. We also generate follow-up questions (incremental and conterfactual) in standards where those apply, yielding 9607 total problems. We evaluate a suite of 29 language models, both proprietary and open. Our analysis uncovers surprising failures, particularly in response to simple follow-up questions, revealing notable gaps even in strong models. Moreover, to the best of our knowledge, we perform the first analysis of the learning dynamics of mathematical skills during LLM training, leveraging checkpoints of Pythia 12B (Biderman et al., 2023). Our contributions are:

- We present MathCAMPS, a framework for synthesizing high-quality mathematical word problems at scale, stratified into fine-grained capabilities defined by the Mathematics Common Core Standards for K-8 grades. We release 9607 problems and our extensible pipeline to generate arbitrarily many more.

- We introduce a cycle-consistency method to validate the faithfulness of the generated word problems to their underlying symbolic structures.

- We propose a novel task of "mathematical dialogue," featuring counterfactual and incremental follow-up questions that probe the models' understanding more deeply.

- We evaluate a diverse set of 29 language models on our dataset, revealing surprising failures and gaps in performance, even in strong models.

## 2 RELATED WORK

Our work closely relates to (i) current benchmarks of mathematical reasoning in LLMs, (ii) benchmarks constructed using LLMs, and (iii) behavioral testing and applications in NLP.

**Benchmarks of mathematical reasoning** MATH (Hendrycks et al., 2021) and GSM8K (Cobbe et al., 2021) have been two leading benchmarks for the evaluation of mathematical reasoning in LLMs. Both datasets consist entirely of human-authored problems — a process that is expensive to reproduce, and as a result, neither benchmarks were updated since their initial releases. Given that LLMs are trained on Web data, it is unclear whether they might have been trained on the test problems of these benchmarks (Bubeck et al., 2023) – either directly or from other sources (e.g., all problems in MATH come from past public competitions). In fact, GSM1K (Zhang et al., 2024), a new dataset that independently attempted to reproduce the data distribution of GSM8K, has found reduced performance on several models, suggesting test set contamination.

**LLM-generated synthetic datasets for LLMs** As collecting data from human annotaotors at scale is expensive (especially in domains requiring expertise, such as mathematics), prior work has relied on LLMs to aid the generation of large-scale benchmarks (Hartvigsen et al., 2022). BigToM (Gandhi et al., 2023), a benchmark of social reasoning in LLMs, applied the idea of symbolically scaffolding questions for the LLM to realize in natural language, an approach that we transport to mathematics. Dyval (Zhu et al., 2024) proposed a method for generating reasoning problems for LLMs based on a DAG representing the computation. While Dyval contains two mathematical tasks (arithmetic and solving linear equations), MathCAMPS takes this idea further for mathematical reasoning, spanning 44 skills directly grounded on a human curriculum. Other synthetic evaluations focused on mathematical skills include GSMore (Hong et al., 2024) and the concurrent work on GSM-Symbolic (Mirzadeh et al., 2024). Both these works focus on evaluating the robustness of LLMs by *perturbing* existing problems from an existing dataset, GSM8k, whereas in MathCAMPS we synthesize problems from scratch, grounded on a human curriculum (Hong et al. (2024) also proposes perturbations to coding problems, which we do not focus on here).

**Behavioral testing in NLP** Our goal to provide a fine-grained evaluation of mathematical reasoning has parallels with *behavioral testing* — the idea of testing software systems on specific features, as opposed to just their overall adequacy (Ribeiro et al., 2020). In particular, CheckList (Ribeiro et al., 2020) allowed testing machine translation models for fine-grained failure modes. Dynaboard (Ma et al., 2021) proposed an NLP leaderboard where users can adapt to their own needs by choosing the utility of different metrics; our dataset enables a similar user-customizable comparison between models for mathematical reasoning.

## 3 MATHCAMPS

We now describe our pipeline for automatically generating mathematical problems and follow-up questions that are grounded in a human curriculum – the Mathematics Common Core (`https://www.thecorestandards.org`). Figure 1 overviews our pipeline. We describe the Common Core, how we represent its standards in a grammar, sample symbolic problems, generate follow-ups, realize those in natural language, and finally improve quality by checking for cycle consistency.

### 3.1 THE MATHEMATICS COMMON CORE

To ground problems in a human curriculum, we turn to the Common Core State Standards for Mathematics. 41 states in the United States adopt the CC as their curriculum. The CC details the mathematical content that students should master from Kindergarten up to 12th grade. Within each grade, the CC elaborates a series of individual *standards*, which detail a particular mathematical skill that students should learn at that grade. Each standard has an identifier, such as `K.CC.C.7`, and a summary description — for `K.CC.C.7`, this is "Compare two numbers between 1 and 10 presented as written numerals". Here, `K` indicates that this is a standard for the Kindergarten grade level, whereas `8.EE.C.8` — "Analyze and solve pairs of simultaneous linear equations" — is an 8th grade standard.

We take 44 standards spanning grades K through 8 to compose MathCAMPS, focusing on standards that are amenable to automatic problem generation with a final answer in text form. The complete CC curriculum has 229 standards across grades K through 8, bring our coverage to 19.2% of the curriculum for these grades. Notably, we currently do not cover standards focusing on conceptual understanding (e.g., `3.OA.D.9` – "Identify arithmetic patterns [...], and explain them using properties of operations."), or standards that emphasize visual reasoning (e.g., `6.G.A.4` – "Represent three-dimensional figures using nets made up of rectangles and triangles, and use the nets to find the surface area of these figures."). All 44 standards covered in MathCAMPS are listed in Appendix A.

**Representing Common Core standards**  We represent CC standards as non-terminals in an *attribute grammar* (Heine & Kuteva, 2007) — a rich formalism that can encode semantic, context-sensitive rules. Attribute grammars can encode syntax much like a context-free grammar, but also allow us to embed information processing (e.g., setting and testing conditions on attributes, such as bounds on constants) in the production rules. We map each standard $s$ to a non-terminal $P_s$, such that all strings produced by expanding $P_s$ using production rules are valid symbolic representations of a problem pertaining to standard $i$. Figure 1 shows a (simplified) grammar for the standard `1.OA.A.1` – "Use addition and subtraction within 20 to solve word problems involving situations of adding to, taking from, putting together". Here, a word problem, generated by the `Problem` non-terminal, consists of a *sequence* of declarative statements expressing equations between expressions. For this standard, an expression consists of addition, subtraction, variables, and constants. After these declarations, the problem ends with a *question* — an expression representing the value that the problem asks for. Concretely, our grammar is implemented in Python: each non-terminal becomes a stochastic function that samples and applies a production rule, recursively expanding non-terminals that it produces. In the grammar in Figure 1 (A), sampling a `Problem` generates a structure such as the one shown in Figure 1 (B).

**Enforcing problem constraints**  When sampling problems, there is no a priori guarantee that all generated statements are necessary to answer the question. To avoid such statements, we remove them by applying a simple graph reachability algorithm on a dependency graph between statements, removing statements that the answer does not depend on. This enforces the constraint of only having useful statements in problems. Besides this constraint, which we always enforce, each standard can apply specific constraints. The standard `1.OA.A.1` has an example of such constraint: it requires that students only be asked to use "addition and subtraction within 20." To be faithful to this standard, we must validate that no intermediate values used in the solution exceed 20. To encode this and other constraints across the curriculum, we implement a suite of 6 parameterized filters (detailed in Appendix C) that are selectively applied depending on the standard's specification. Applying rejection sampling from the grammar using the standard's filters gives a procedure for generating valid symbolic *problems*. For all standards that can be formulated as solving a system of linear

equations, we use SymPy (Meurer et al., 2017) to obtain final answers. For other cases, we use two simple custom procedures (to list the factors of numbers and to compare values).

## 3.2 FROM SYMBOLIC TO WORD PROBLEMS

To realize the symbolic problems into natural language, we use few-shot prompting with GPT-4 (Figure 1 (C)). For each standard, we sampled two valid symbolic problems and manually wrote a problem in natural language that faithfully represents the symbolic structure. For standards involving word problems, which typically contain a simple cover story, we also sampled a random theme out of 188 that we crafted (e.g., "Book", "Pirate ship", "Money"). These examples are then given to GPT-4 in-context, along with a new symbolic structure (and a random theme, for standards where that is relevant), requesting it to generate a faithful natural language problem for that structure.

Unlike generating problem stories from a fixed set of templates, using a language model for generating natural language problems gives us fluid, diverse language. Unfortunately, we also lose any guarantee that the generated word problem represents the original symbolic structure faithfully. To mitigate this issue, we also introduce a *cycle consistency* method that we have found to drastically improve problem quality. Precisely, we use the same few-shot examples we crafted for each standard *in reverse* (i.e., with the natural language problem coming first, followed by the symbolic structure) to have GPT-4 translate the word problem it wrote into a symbolic structure. In this step, the model is not given the original structure. We then parse and apply the appropriate solver to the generated symbolic problem; we consider the generation *cycle-consistent* if the answers to the original and recovered problems are the same (illustrated in Figure 1). We then discard problems that fail this test.

This cycle consistency test significantly improves the reliability of our pipeline. We manually evaluated 245 random problems generated by sampling a symbolic structure and then a word problem from GPT-4. Out of those, we identified 30 word problems (12.2%) that were not faithful to the original symbolic structure — for those, the answer that we compute to the *symbolic problem* does not match our manual solution to the *word problem*. Cycle consistency discarded 25 of those (and 7 problems that were indeed faithful). Out of the remaining 215 problems, 210 (97.7%) were judged as faithful in our manual check. A more in-depth analysis of cycle-consistency can be found in Appendix D.

## 3.3 GENERATING FOLLOW-UP QUESTIONS

As human instructors know, follow-up questions are often the best way to probe a student's understanding. In MathCAMPS, we leverage our symbolic representation of problems to derive follow-up questions. We propose two kinds of questions: *counterfactual* questions, where we change a constant in the original problem, and *incremental* questions, where we add a new piece of information. For each CC standard, we mark which (if any) of these two categories of follow-ups are applicable. Symbolically, follow-up questions are represented as a *difference* to be applied to the original question — when we apply the difference, we obtain a new problem. We then use the same solver as the original problem to obtain the ground-truth answer to the follow-up question. We employ the same few-shot structure to translate this difference into a natural language question, and parse it back into a symbolic structure to test for cycle consistency.

## 4 EXPERIMENTS

We now evaluate a suite of 29 LLMs from 11 different vendors on MathCAMPS. We evaluate all models by sampling with temperature 0, using a fixed 1-shot prompt with the first example from GSM8K, mostly to demonstrate the format. For all models (most of them instruction-tuned), a single example was enough for to adhere to the task and the format we specify. The rich structure in MathCAMPS allows us to perform a number of unique analyses on LLMs relating to specific mathematical abilities and their corresponding grade levels for human students. Precisely, we investigate:

1. How do LLMs perform overall on MathCAMPS? How does their performance correlate with GSM8k?

Table 1: Final answer accuracy of LLMs on MathCAMPS, both over all problems (**All**) and considering only standards in each grade we cover (**K** to **8**). Highlights compare to gradewise avg.

| Vendor | Model | All | K | 1 | 2 | 3 | 4 | 5 | 6 | 7 | 8 |
|--------|-------|-----|---|---|---|---|---|---|---|---|---|
| OpenAI | GPT-4o | 0.92 | 0.98 | 0.98 | 0.98 | 0.98 | 0.92 | 0.88 | 0.95 | 0.89 | 0.64 |
| Anthropic | Claude-3 Opus | 0.89 | 0.97 | 0.99 | 0.96 | 0.98 | 0.89 | 0.83 | 0.96 | 0.73 | 0.56 |
| Google | Gemini-1.5 Pro | 0.89 | 0.95 | 0.98 | 0.97 | 0.97 | 0.89 | 0.83 | 0.93 | 0.78 | 0.54 |
| Google | Gemini-1.5 Flash | 0.87 | 0.98 | 0.98 | 0.97 | 0.98 | 0.80 | 0.80 | 0.90 | 0.84 | 0.56 |
| OpenAI | GPT-3.5 Turbo | 0.87 | 0.96 | 0.98 | 0.98 | 0.97 | 0.86 | 0.77 | 0.90 | 0.77 | 0.56 |
| Anthropic | Claude-3 Sonnet | 0.86 | 0.96 | 0.98 | 0.97 | 0.98 | 0.88 | 0.74 | 0.94 | 0.66 | 0.49 |
| Anthropic | Claude-3 Haiku | 0.84 | 0.97 | 0.98 | 0.97 | 0.98 | 0.87 | 0.69 | 0.92 | 0.59 | 0.51 |
| Qwen | Qwen2-Math 72B | 0.89 | 0.98 | 0.99 | 0.98 | 0.97 | 0.90 | 0.80 | 0.91 | 0.77 | 0.59 |
| Meta | Llama 3 70B | 0.85 | 0.96 | 0.97 | 0.97 | 0.97 | 0.85 | 0.71 | 0.87 | 0.73 | 0.50 |
| Mistral | Mixtral 8x22B | 0.84 | 0.96 | 0.99 | 0.98 | 0.96 | 0.79 | 0.69 | 0.88 | 0.73 | 0.61 |
| Qwen | Qwen2-Math 7B | 0.83 | 0.96 | 0.99 | 0.97 | 0.93 | 0.85 | 0.66 | 0.91 | 0.58 | 0.62 |
| DeepSeek | DeepSeek 67B | 0.80 | 0.95 | 0.99 | 0.96 | 0.93 | 0.82 | 0.60 | 0.84 | 0.61 | 0.47 |
| DeepSeek | DeepSeek Math 7B Base | 0.78 | 0.94 | 0.97 | 0.93 | 0.89 | 0.75 | 0.63 | 0.86 | 0.53 | 0.55 |
| Numina | NuminaMath 7B TIR | 0.78 | 0.89 | 0.97 | 0.95 | 0.90 | 0.72 | 0.63 | 0.84 | 0.59 | 0.53 |
| Meta | Llama 3 8B | 0.77 | 0.94 | 0.97 | 0.96 | 0.94 | 0.78 | 0.55 | 0.79 | 0.53 | 0.43 |
| Mistral | Mixtral 8x7B | 0.76 | 0.94 | 0.96 | 0.93 | 0.91 | 0.75 | 0.52 | 0.80 | 0.53 | 0.45 |
| InternLM | InternLM-Math Base 20B | 0.74 | 0.95 | 0.96 | 0.95 | 0.86 | 0.68 | 0.55 | 0.79 | 0.52 | 0.47 |
| EleutherAI | Llemma 34B | 0.71 | 0.95 | 0.96 | 0.93 | 0.87 | 0.61 | 0.47 | 0.77 | 0.46 | 0.44 |
| Mistral | Mistral 7B | 0.68 | 0.89 | 0.94 | 0.91 | 0.84 | 0.61 | 0.42 | 0.66 | 0.45 | 0.42 |
| DeepSeek | DeepSeek Coder 33B | 0.65 | 0.88 | 0.93 | 0.92 | 0.83 | 0.54 | 0.36 | 0.66 | 0.44 | 0.38 |
| Meta | CodeLlama 34B | 0.64 | 0.90 | 0.94 | 0.92 | 0.85 | 0.51 | 0.38 | 0.70 | 0.37 | 0.30 |
| Microsoft | phi-2 | 0.63 | 0.95 | 0.96 | 0.89 | 0.78 | 0.46 | 0.38 | 0.61 | 0.37 | 0.41 |
| EleutherAI | Llemma 7B | 0.62 | 0.78 | 0.90 | 0.85 | 0.79 | 0.48 | 0.41 | 0.67 | 0.41 | 0.36 |
| Google | Gemma 7B | 0.62 | 0.83 | 0.92 | 0.90 | 0.82 | 0.47 | 0.36 | 0.65 | 0.36 | 0.30 |
| Meta | CodeLlama 13B | 0.58 | 0.87 | 0.92 | 0.87 | 0.75 | 0.41 | 0.30 | 0.61 | 0.32 | 0.34 |
| InternLM | InternLM-Math Base 7B | 0.58 | 0.71 | 0.73 | 0.73 | 0.72 | 0.54 | 0.38 | 0.61 | 0.37 | 0.39 |
| Meta | CodeLlama 7B | 0.52 | 0.85 | 0.92 | 0.84 | 0.69 | 0.37 | 0.25 | 0.57 | 0.25 | 0.16 |
| Google | Gemma 2B | 0.51 | 0.66 | 0.76 | 0.74 | 0.67 | 0.42 | 0.28 | 0.55 | 0.30 | 0.27 |
| - | Avg. Performance | 0.75 | 0.91 | 0.95 | 0.92 | 0.88 | 0.70 | 0.57 | 0.79 | 0.56 | 0.46 |

2. Do individual models have relative strengths and weaknesses, or does performance improve uniformly across skills?

3. How well do LLMs respond to follow-up questions? How is their accuracy affected when also considering follow-ups?

4. How do mathematical skills develop during pre-training?

## 4.1 OVERALL PERFORMANCE

Table 1 shows both aggregate accuracy on MathCAMPS, as well as accuracy across standards partitioned by grade, whereas Figure 3 compares the aggregate accuracies on MathCAMPS and GSM8K. Closed-weights models are shown above the line, with open-weights models below. GPT-4o ranks at the top in overall accuracy. Since we used GPT-4 to generate the problems, we must rule out familiarity bias (Stureborg et al., 2024) in this result. We thus generated a 10%-scale dataset with the same pipeline but using Claude-3 Opus. We found that GPT-4o still outperforms Claude-3 Opus on this dataset (see Appendix B), suggesting that its advantage on MathCAMPS was not due to a familiarity bias. We make the following observations:

**Models of similar overall performance can have large disparities in specific abilities or grades.** Several models that have comparable overall accuracies show large differences when compared on specific mathematical skills. As an example, Claude-3 Opus and Claude-3 Sonnet have similar overall accuracy both in MathCAMPS (.89 vs .86) and in GSM8K (.95 vs .923). However, we find that Claude-3 Opus is significantly better at manipulating fractions. For instance, in the CC standard 5.NF.A.2, described as *"Solve word problems involving addition and subtraction of fractions referring to the same whole, including cases of unlike denominators"*, Opus has a 36%

Table 2: Largest model rank changes when focusing on one CC standard. Here, A ↗B indicates that the model ranks $A^{th}$ on MathCAMPS overall, but ranks $B^{th}$ when only evaluating on problems from the indicated CC standard. Conversely, ↘marks notable cases where a model's performance on the indicated CC standard is lower than its overall performance on MathCAMPS. We show selected rows here, the complete table can be found in the Appendix.

| Model | Top outlier skill | Rank change |
|---|---|---|
| GPT-4o | 8.EE.C.8 - Solve two-variable systems | $(1^{st} ↘22^{th})$ |
| Claude-3 Opus | 2.MD.B.5 - Add/sub within 100 | $(2^{nd} ↘18^{th})$ |
| Gemini-1.5 Pro | K.OA.A.4 - Adding to equal 10 | $(4^{th} ↘23^{th})$ |
| Claude-3 Haiku | 6.EE.A.1 - Evaluate exponents | $(10^{th} ↘20^{th})$ |
| Llama 3 70B | 3.OA.A.3 - Mul/div within 100 | $(8^{th} ↘21^{th})$ |
| Mixtral 8x22B | 8.EE.C.8 - Solve two-variable systems | $(9^{th} ↘21^{th})$ |
| Qwen2-Math 7B | 8.EE.C.8 - Solve two-variable systems | $(11^{th} ↘25^{th})$ |
| DeepSeek 67B | K.NBT.A.1 - Decompose into 10s | $(12^{th} ↗1^{st})$ |
| Llama 3 8B | K.OA.A.4 - Adding to equal 10 | $(15^{th} ↗3^{rd})$ |
| Mixtral 8x7B | 6.EE.A.1 - Evaluate exponents | $(16^{th} ↘26^{th})$ |
| InternLM-Math Base 20B | 2.NBT.B.5 - Add/sub within 100 | $(17^{th} ↗2^{nd})$ |
| Llemma 34B | 3.OA.A.3 - Mul/div within 100 | $(18^{th} ↗1^{st})$ |
| Mistral 7B | 1.OA.A.1 - Add/sub within 20 | $(19^{th} ↘26^{th})$ |
| DeepSeek Coder 33B | 6.EE.A.1 - Evaluate exponents | $(20^{th} ↗3^{rd})$ |
| phi-2 | K.OA.A.4 - Adding to equal 10 | $(22^{th} ↗4^{th})$ |
| Llemma 7B | 6.EE.A.1 - Evaluate exponents | $(23^{th} ↗5^{th})$ |
| Gemma 7B | K.OA.A.5 - Add/sub within 5 | $(24^{th} ↗6^{th})$ |
| InternLM-Math Base 7B | 4.OA.B.4 - Factor pairs within 100 | $(26^{th} ↗15^{th})$ |
| CodeLlama 7B | 8.EE.C.8 - Solve two-variable systems | $(27^{th} ↗15^{th})$ |
| Gemma 2B | 8.EE.C.8 - Solve two-variable systems | $(28^{th} ↗11^{th})$ |

advantage over Sonnet, scoring a 70% accuracy for this standard, whereas Sonnet only achieves 34%. Similarly, while Gemma 7B and phi-2 have comparable overall performance (.62 vs .63 accuracy on MathCAMPS), some capabilities in each model seem nearly absent from the other. Gemma 7B is highly accurate when performing multi-digit multiplication — an ability stressed in standard 4.NBT.B.4, where Gemma 7B achieves 94% accuracy. In stark contrast, phi-2 only solves 22% of those problems. On the other direction, phi-2 is one of the highest performing models on 4.NF.A.2 ("Compare two fractions with different numerators and different denominators"), with 90% accuracy. In this same standard, Gemma 7B only scores 19%. Such stark differences are obscured when only analyzing aggregate metrics, whereas MathCAMPS allows for a much more nuanced understanding of mathematical reasoning capabilities.

**Overall ranking between models is largely a function of which skills we choose to evaluate.** Overall accuracies in any dataset induce a single performance ranking of models. However, when we look at individual CC standards in MathCAMPS, rankings are largely a function of which skills we choose to evaluate. Comparing pairs of models across all standards, rarely we find cases where one model Pareto-dominates another (i.e. is better on all standards): only 23.08% of all pairs of models have a Pareto winner. Table 2 shows how the ranking of a model in individual skills can often deviate strongly from its overall ranking. Here, the first ordinal in each cell shows the model's global ranking when comparing overall performance in MathCAMPS, whereas the second shows the model's ranking on that particular CC standard. We find many cases of large discrepancies. For instance, on systems of equations, GPT-4o tends to excessively rely on decimal approximations when operating with fractions, resulting in poor performance. Llemma 34B, which places 13th overall, is the best performing model on a simple kindergarten-level word problems on adding to complete 10.

**Aggregate accuracies are strongly correlated between GSM8k and MathCAMPS** When considering overall performance, the trends in GSM8k hold on the novel problems from MathCAMPS, which cover overlapping topics (Pearson correlation of 0.865, $p < 10^{-5}$; we show this correlation in Figure 3). This correlation corroborates the progress that public benchmarks have witnessed, suggesting that data contamination does not play a major role in explaining observed improvements

Table 3: Standards with strict winners, i.e., models who strictly outperform all other models on that standard.

| Model | Standards Won |
|---|---|
| GPT-4o | 4.NBT.B.6, 7.NS.A.2, 8.EE.C.7, 7.NS.A.1-fraction, 5.NF.A.1, 7.NS.A.3-fraction |
| Qwen2-Math 72B | 1.OA.A.1, 3.OA.D.8, 5.NF.B.4, 4.OA.A.3, 4.MD.A.2-fraction |
| GPT-3.5 Turbo | 2.NBT.B.6, 5.OA.A.1, 8.EE.C.8 |
| Claude-3 Opus | 6.NS.B.2, 5.NBT.B.7 |
| Gemini-1.5 Flash | 7.NS.A.3-decimal, 5.NF.A.2 |
| Claude-3 Sonnet | 3.MD.D.8-polygon |

in recent LLMs. We note that prior work attempting to replicate the distribution of GSM8k, such as the independent effort to collect GSM1k (Zhang et al., 2024), has observed a smaller correlation, including substantial drops in performance for some models. This is entirely compatible with our findings here, due to the difficulty of exactly replicating the distribution over skills in any given human-created benchmark. As the sharp differences in Table 2 indicate, an (unintended) shift in this distibution can drastically — and unevenly — affect accuracy, even if no data contamination occurs. These shifts are easily avoided in an automated pipeline as in MathCAMPS, allowing us to draw new problems from the exact same distribution in the future.

## 4.2 STANDARD-SPECIFIC ANALYSIS

Despite decently high performance across the board, GPT-4o's performance fell at or below $90\%$ on the following skills: 4.MD.A.2-fraction, 4.OA.A.3, 5.NF.A.1, 7.NS.A.3-fraction, and 8.EE.C.8. At their core, all these abilities require fraction addition or subtraction, a skill we noted that GPT-4o struggles with. Specifically, the model starts approximating fractions using decimals, and the error introduced by this compounds throughout the problem, resulting in an incorrect final answer. Surprisingly, GPT-4o achieves an $86\%$ on 5.NF.B.4, which requires fraction multiplication, indicating that it is likely the multi-step process of finding common denominators in adding/subtracting fractions that challenges GPT-4o. Additionally, GPT-4o achieves performances above $90\%$ on 4.MD.A.2-decimal and 7.NS.A.3-decimal, which are the CC standards equivalent to 4.MD.A.2-fraction and 7.NS.A.3-fraction, using decimals instead of fractions in the problems. This trend isn't isolated to the GPT models, though, as most models tended to struggle more with standards involving fractions.

Work from Lucy et al. (2024) showed that over $50\%$ of problems from GSM8K originated from three CC standards, namely, 4.OA.A.3 ($20.73\%$), 2.OA.A.1 ($16.58\%$), and 3.OA.D.8 ($15.75\%$). These standards ask students to solve multistep word problems involving the four operations, use addition and subtraction to solve two-step word problems within 100, and solve two-step word problems using the four operations, respectively. While most models we experimented with performed relatively well on 2.OA.A.1 and 3.OA.D.8, CC standard 4.OA.A.3 did prove to be challenging, with the most performant model, Qwen2-Math 72B, achieving an $86\%$ on the standard.

Out of the 49 total skills we evaluated (44 standards, some of which we split into sub-standards), 19 skills had an absolute winner: a model which outperforms all other models on that skill. The distribution of these skills is given in Table 3. This analysis shows that even generally weaker models, such as GPT-3.5 Turbo, have particular skills that they excel on. This fact *is hidden when looking at aggregate accuracies*, but is revealed in our finer-grained analysis.

## 4.3 FOLLOW-UP TASKS

We now evaluate the performance of language models when asked follow-up questions. Here, we first give the initial problem, and in case the model answers correctly we ask either an incremental follow-up, a counterfactual follow-up, or both (in separate contexts), depending on the standard (some standards don't have follow-ups, and for some problems we failed to find a cycle-consistent follow-up within the max attempts). Here, we're interested in analyzing the (lack of) robustness that LMs might have when probed with extra questions — our follow-ups are generally answerable using the same core mathematical knowledge involved in the initial problem but require longer range attention and dialog understanding.

Table 4: Model performance on our mathematical dialogue task, where the model must answer follow-up questions besides the initial problem. The second column, **Acc**uracy **with follow-ups**, shows overall success rate across standards that contain follow-up questions, considering a model successful only when it answers a problem and its follow-up questions correctly. The third and fourth columns show the hardest standard for each model when it comes to follow-up questions, showing a standard's code and abbreviated description, the model's accuracy ignoring follow-ups, and after follow-ups. We show selected rows here, the complete table can be found in the Appendix.

| Model | Acc. with follow-ups | Largest accuracy drop w/ follow-ups | |
|---|---|---|---|
| GPT-4o | 0.82 | 5.NF.A.1 - Add/sub fractions | 0.86 ↘0.58) |
| Claude-3 Opus | 0.76 | 7.NS.A.1-fraction - Add/sub with fractions | 0.54 ↘0.23) |
| Gemini-1.5 Pro | 0.77 | 5.OA.A.1 - Evaluating with parentheses | 0.95 ↘0.69) |
| Claude-3 Haiku | 0.70 | 7.NS.A.2 - Mult/div with fractions | 0.55 ↘0.26) |
| Qwen2-Math 72B | 0.78 | 5.NF.A.1 - Add/sub fractions | 0.49 ↘0.23) |
| Llama 3 70B | 0.69 | 4.NF.A.2 - Compare two fractions | 0.99 ↘0.66) |
| Mixtral 8x22B | 0.69 | 7.NS.A.1-fraction - Add/sub with fractions | 0.69 ↘0.17) |
| Qwen2-Math 7B | 0.71 | 5.NF.A.2 - Add/sub fraction word problems | 0.41 ↘0.17) |
| DeepSeek Math 7B Base | 0.65 | 5.NF.B.4 - Mult fractions | 0.81 ↘0.57) |
| NuminaMath 7B TIR | 0.62 | 5.NF.A.2 - Add/sub fraction word problems | 0.44 ↘0.18) |
| Llama 3 8B | 0.58 | 4.NF.A.2 - Compare two fractions | 0.90 ↘0.52) |
| Mixtral 8x7B | 0.58 | 7.NS.A.2 - Mult/div with fractions | 0.60 ↘0.28) |
| Llemma 34B | 0.55 | 5.NF.B.4 - Mult fractions | 0.68 ↘0.31) |
| Mistral 7B | 0.48 | 7.NS.A.1-decimal - Add/sub with decimals | 0.91 ↘0.50) |
| DeepSeek Coder 33B | 0.60 | 3.OA.A.3 - Mul/div within 100 | 0.95 ↘0.81) |
| phi-2 | 0.39 | 3.NBT.A.2 - Add/sub within 1000 | 0.71 ↘0.23) |
| Llemma 7B | 0.42 | 5.NF.B.4 - Mult fractions | 0.58 ↘0.21) |
| Gemma 7B | 0.33 | 7.NS.A.1-decimal - Add/sub with decimals | 0.91 ↘0.32) |
| InternLM-Math Base 7B | 0.42 | 7.NS.A.1-decimal - Add/sub with decimals | 0.82 ↘0.47) |
| CodeLlama 7B | 0.49 | 2.NBT.B.7 - Add/sub within 100 | 0.80 ↘0.67) |
| Gemma 2B | 0.24 | 3.NBT.A.2 - Add/sub within 1000 | 0.93 ↘0.26) |

Table 4 shows overall accuracies when we only consider a model successful on a problem when it also answers its follow-up questions correctly (the full table, with results for all models, is given in the Appendix; see Table18). We also show the major accuracy drops across CC standards for each model (last two columns). We find many notable cases, in both stronger and weaker models. GPT-4o, for instance, is 90% accurate in evaluating expressions of addition of fractions with multi-digit numerators and denominators (5.NF.A.1 — notably, this requires putting fractions in the same denominator). When asked to add another fraction to the result, or change one of the original fractions to a new one and re-do the computation, its success rate when evaluated at correctly answering both follow-ups drops to 61%, or a 29% decrease. Other models drop even more dramatically. For instance, phi-2 solves 57% of the problems in 7.NS.A.2, which are about multiplying two fractions (only requires two multi-digit multiplications — we do not require the result to be in lowest terms). However, when asked to multiply the result by a further third fraction, phi-2 tends to not reuse its previous (correct) result, and instead proceeds by writing down the product of the three numerators (and denominators), and attempt to directly evaluate this product. This strategy is rarely successful, and it only achieves 8% accuracy when accounting for the follow-ups (an absolute 49% drop). Overall, we find many cases where models are not robust to simple follow-up questions. We hypothesize that this setup of mathematical dialogue is much less frequent in pre-training data, and that follow-up problems in MathCAMPS can be a rich source of further analyses for future work.

## 4.4 LEARNING DYNAMICS

Finally, we use Pythia (Biderman et al., 2023) to showcase another analysis that MathCAMPS enables: understanding the learning dynamics of mathematical skills during LM training. We evaluate checkpoints of Pythia 12B on all standards, and track the performance change as the model was trained. Figure 2 shows Pythia's performance evolving during training on all 7 CC standards where the last checkpoint achieves at least 30% accuracy. Early in training, after 28k steps, Pythia performs best in a Kindergarten standard, K.OA.A.5 — "Fluently add and subtract within 5.". At 57k steps, its performance is best in both K.OA.A.5 (37% accuracy) and two first-

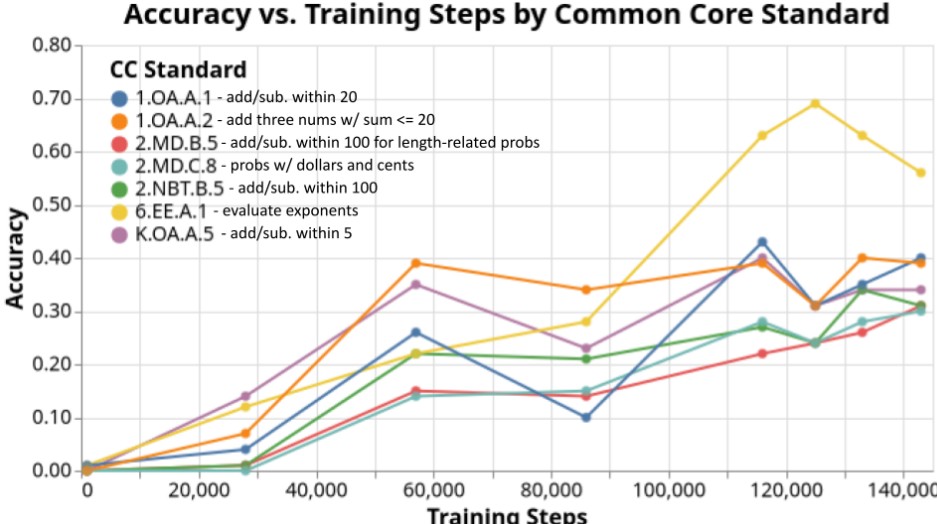

Figure 2: Performance of Pythia 12B checkpoints on MathCAMPS standards as it evolves during training. We show all 7 standards where the last checkpoint has at least 30% accuracy.

grade standards, 1.OA.A.1 and 1.OA.A.2 — both standards involve simple word problems with addition and subtraction within 20. Pythia starts to become proficient at a sixth-grade standard around midway during training: 6.EE.A.1, which involves evaluating simple expressions using whole-number exponents (e.g, computing squares and cubes). These skills develop in tandem with its linguistic competence – at first, Pythia repeats questions verbatim often, but at 57k steps it already often produces *responses*. Overall, the high-resolution of MathCAMPS as a reasoning benchmark can support future work to deepen our understanding of how language models acquire capabilities during training, and how specific factors (such as data, or scale) contribute to their learning.

## 5    CONCLUSION

We introduce MathCAMPS, a fine-grained synthetic benchmark of mathematical reasoning in LLMs. MathCAMPS is directly grounded on the Common Core Standards, a widely used curriculum in human education. By tying our problems to a human curriculum, we enable a much wider range of analyses to understand mathematical reasoning capabilities and weaknesses of LLMs. We show analyses of performance by grade level and identify particularly challenging skills for a range of models, though we believe these are only a few examples of analyses that MathCAMPS permits.

We note that MathCAMPS might also find applications in educational tools for human students, due to its correspondence to the Common Core. Future work in that direction will require psychometric analyses, to ensure that problem difficulty (aside from the abilities involved) is grade appropriate.

While we currently cover 44 CC standards, our pipeline can be easily extended to cover additional standards where problems have a computational nature, and where answers can be obtained using a computer solver. These can include topics beyond high-school, including calculus and linear algebra. This framework, however, is difficult to extend to more conceptual problems, including mathematical proofs, or problems that require *explanations*, as opposed to a final computational answer. Judging natural language reasoning reliably, in the absence of an exact answer to compare to, remains an open problem — an important challenge to allow us to extend the scope of evaluation of mathematical reasoning in LLMs.

**Reproducibility Statement**    MathCAMPS is a fully synthetic dataset, and we have made available both the code to run our full dataset generation pipeline (available in the supplementary materials) as well as our analyses with the existing problems and collected LLM responses (`analysis.py`, available in the supplementary materials, along with the JSON data files under `model-responses`, containing all LLM-generated solutions to the problems). The problems we generated for this paper

came from GPT-4, a closed model, and its availability is subject to change. However, our pipeline still works with other models: our analysis in Appendix B shows that using Claude leads to a dataset with similar results. We thus expect our pipeline to be reproducible with other strong models that are available, including open ones.

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

| Standard ID | Description |
|---|---|
| K.CC.C.7 | Compare two numbers between 1 and 10 presented as written numerals. |
| K.OA.A.4 | For any number from 1 to 9, find the number that makes 10 when added to the given number, e.g., by using objects or drawings, and record the answer with a drawing or equation. |
| K.OA.A.5 | Fluently add and subtract within 5. |
| K.NBT.A.1 | Compose and decompose numbers from 11 to 19 Into ten ones and some further ones, e.g., by using objects or drawings, and record each composition or decomposition by a drawing or equation (e.g., 18 = 10 + 8); understand that these numbers are composed of ten ones and one, two, three, four, five, six, seven, eight, or nine ones. |

Table 5: CC Standards for Grade K

| Standard ID | Description |
|---|---|
| 1.OA.A.1 | Use addition and subtraction within 20 to solve word problems involving situations of adding to, taking from, putting together, taking apart, and comparing, with unknowns in all positions, e.g., by using objects, drawings, and equations with a symbol for the unknown number to represent the problem. |
| 1.OA.A.2 | Solve word problems that call for addition of three whole numbers whose sum is less than or equal to 20, e.g., by using objects, drawings, and equations with a symbol for the unknown number to represent the problem. |
| 1.OA.D.8 | Determine the unknown whole number in an addition or subtraction equation relating three whole numbers. |

Table 6: CC Standards for Grade 1

## A    COMMON CORE STANDARDS IN MATHCAMPS

MathCAMPS is available on Github at `https://github.com/<<redacted>>/mathcamps`. All of the Common Core standards we implement are described in a configuration file, `commoncore.yaml`, where standards are instantiated by composing high-level components from the Common Core attribute grammar. Moreover, we provide our prompts used to generate the dataset and model responses, as well as all problems and model responses for all LLMs we evaluated.

We list the Common Core standards we represent in MathCAMPS in Tables 5 through 13, segregated by grade. Standards 3.MD.D.8, 4.MD.A.2, 7.NS.A.1, and 7.NS.A.3 are split up into sub-standards. This was done for ease of implementation of the grammar.

## B    FAMILIARITY BIAS

MathCAMPS was generated using GPT-4. GPT-4o, a model of the same family, was also the best performer overall (Table 1). To test whether this might be due to a familiarity bias — problems being in-distribution for GPT-4o, but out-of-distribution for other models —, we generated a 10%-scale dataset using the exact same pipeline, but using Claude 3 Opus for both generating word problems and testing cycle consistency. This dataset has the same distribution of standards as MathCAMPS. We evaluated GPT-4o and Claude 3 Opus on this dataset — accuracies are reported in Table 14. GPT-4o also performs better in this dataset, suggesting that its performance in MathCAMPS was not due to a higher relative familiarity with the problems.

| Standard ID | Description |
|---|---|
| 2.OA.A.1 | Use addition and subtraction within 100 to solve one- and two-step word problems involving situations of adding to, taking from, putting together, taking apart, and comparing, with unknowns in all positions, e.g., by using drawings and equations with a symbol for the unknown number to represent the problem. |
| 2.NBT.B.5 | Fluently add and subtract within 100 using strategies based on place value, properties of operations, and/or the relationship between addition and subtraction. |
| 2.NBT.B.6 | Add up to four two-digit numbers using strategies based on place value and properties of operations. |
| 2.NBT.B.7 | Add and subtract within 1000, using concrete models or drawings and strategies based on place value, properties of operations, and/or the relationship between addition and subtraction; relate the strategy to a written method. Understand that in adding or subtracting three-digit numbers, one adds or subtracts hundreds and hundreds, tens and tens, ones and ones; and sometimes it is necessary to compose or decompose tens or hundreds. |
| 2.MD.B.5 | Use addition and subtraction within 100 to solve word problems involving lengths that are given in the same units, e.g., by using drawings (such as drawings of rulers) and equations with a symbol for the unknown number to represent the problem. |
| 2.MD.C.8 | Solve word problems involving dollar bills, quarters, dimes, nickels, and pennies, using $ and ¢ symbols appropriately. |

Table 7: CC Standards for Grade 2

| Standard ID | Description |
|---|---|
| 3.OA.A.3 | Use multiplication and division within 100 to solve word problems in situations involving equal groups, arrays, and measurement quantities, e.g., by using drawings and equations with a symbol for the unknown number to represent the problem. |
| 3.OA.A.4 | Determine the unknown whole number in a multiplication or division equation relating three whole numbers. |
| 3.OA.C.7 | Fluently multiply and divide within 100, using strategies such as the relationship between multiplication and division (e.g., knowing that $8 \times 5 = 40$, one knows $40 \div 5 = 8$) or properties of operations. By the end of Grade 3, know from memory all products of two one-digit numbers. |
| 3.OA.D.8 | Solve two-step word problems using the four operations. Represent these problems using equations with a letter standing for the unknown quantity. Assess the reasonableness of answers using mental computation and estimation strategies including rounding. |
| 3.MD.D.8-triangle | Solve real world and mathematical problems involving perimeters of polygons, including finding the perimeter given the side lengths, finding an unknown side length, and exhibiting rectangles with the same perimeter and different areas or with the same area and different perimeters. |
| 3.MD.D.8-quadrilateral | Solve real world and mathematical problems involving perimeters of polygons, including finding the perimeter given the side lengths, finding an unknown side length, and exhibiting rectangles with the same perimeter and different areas or with the same area and different perimeters. |
| 3.MD.D.8-polygon | Solve real world and mathematical problems involving perimeters of polygons, including finding the perimeter given the side lengths, finding an unknown side length, and exhibiting rectangles with the same perimeter and different areas or with the same area and different perimeters. |
| 3.NBT.A.2 | Fluently add and subtract within 1000 using strategies and algorithms based on place value, properties of operations, and/or the relationship between addition and subtraction. |

Table 8: CC Standards for Grade 3

| Standard ID | Description |
|---|---|
| 4.OA.A.3 | Solve multistep word problems posed with whole numbers and having whole-number answers using the four operations, including problems in which remainders must be Interpreted. Represent these problems using equations with a letter standing for the unknown quantity. Assess the reasonableness of answers using mental computation and estimation strategies including rounding. |
| 4.OA.B.4 | Find all factor pairs for a whole number in the range 1-100. Recognize that a whole number is a multiple of each of its factors. Determine whether a given whole number in the range 1-100 is a multiple of a given one-digit number. Determine whether a given whole number in the range 1-100 is prime or composite. |
| 4.NBT.B.4 | Fluently add and subtract multi-digit whole numbers using the standard algorithm. |
| 4.NBT.B.5 | Multiply a whole number of up to four digits by a one-digit whole number, and multiply two two-digit numbers, using strategies based on place value and the properties of operations. Illustrate and explain the calculation by using equations, rectangular arrays, and/or area models. |
| 4.NBT.B.6 | Find whole-number quotients and remainders with up to four-digit dividends and one-digit divisors, using strategies based on place value, the properties of operations, and/or the relationship between multiplication and division. Illustrate and explain the calculation by using equations, rectangular arrays, and/or area models. |
| 4.NF.A.2 | Compare two fractions with different numerators and different denominators, e.g., by creating common denominators or numerators, or by comparing to a benchmark fraction such as 1/2. Recognize that comparisons are valid only when the two fractions refer to the same whole. Record the results of comparisons with symbols ¿, =, or ¡, and justify the conclusions, e.g., by using a visual fraction model. |
| 4.MD.A.2-decimal | Use the four operations to solve word problems involving distances, Intervals of time, liquid volumes, masses of objects, and money, including problems involving simple fractions or decimals, and problems that require expressing measurements given in a larger unit in terms of a smaller unit. Represent measurement quantities using diagrams such as number line diagrams that feature a measurement scale. |
| 4.MD.A.2-fraction | Use the four operations to solve word problems involving distances, Intervals of time, liquid volumes, masses of objects, and money, including problems involving simple fractions or decimals, and problems that require expressing measurements given in a larger unit in terms of a smaller unit. Represent measurement quantities using diagrams such as number line diagrams that feature a measurement scale. |
| 4.MD.A.3 | Apply the area and perimeter formulas for rectangles in real world and mathematical problems. |

Table 9: CC Standards for Grade 4

| Standard ID | Description |
| --- | --- |
| 5.OA.A.1 | Use parentheses, brackets, or braces in numerical expressions, and evaluate expressions with these symbols. |
| 5.NBT.B.5 | Fluently multiply multi-digit whole numbers using the standard algorithm. |
| 5.NBT.B.6 | Find whole-number quotients of whole numbers with up to four-digit dividends and two-digit divisors, using strategies based on place value, the properties of operations, and/or the relationship between multiplication and division. Illustrate and explain the calculation by using equations, rectangular arrays, and/or area models. |
| 5.NBT.B.7 | Add, subtract, multiply, and divide decimals to hundredths, using concrete models or drawings and strategies based on place value, properties of operations, and/or the relationship between addition and subtraction; relate the strategy to a written method and explain the reasoning used. |
| 5.NF.A.1 | Add and subtract fractions with unlike denominators (including mixed numbers) by replacing given fractions with equivalent fractions in such a way as to produce an equivalent sum or difference of fractions with like denominators. |
| 5.NF.A.2 | Solve word problems involving addition and subtraction of fractions referring to the same whole, including cases of unlike denominators, e.g., by using visual fraction models or equations to represent the problem. Use benchmark fractions and number sense of fractions to estimate mentally and assess the reasonableness of answers. |
| 5.NF.B.4 | Apply and extend previous understandings of multiplication to multiply a fraction or whole number by a fraction. |

Table 10: CC Standards for Grade 5

| Standard ID | Description |
| --- | --- |
| 6.NS.B.2 | Fluently divide multi-digit numbers using the standard algorithm. |
| 6.NS.B.3 | Add, subtract, multiply, and divide decimals to hundredths, using concrete models or drawings and strategies based on place value, properties of operations, and/or the relationship between addition and subtraction; relate the strategy to a written method and explain the reasoning used. |
| 6.EE.A.1 | Write and evaluate numerical expressions involving whole-number exponents. |
| 6.EE.B.7 | Solve real-world and mathematical problems by writing and solving equations of the form $x + p = q$ and $px = q$ for cases in which p, q and x are all nonnegative rational numbers. |

Table 11: CC Standards for Grade 6

| Standard ID | Description |
| --- | --- |
| 7.NS.A.1-fraction | Apply and extend previous understandings of addition and subtraction to add and subtract rational numbers; represent addition and subtraction on a horizontal or vertical number line diagram. |
| 7.NS.A.1-decimal | Apply and extend previous understandings of addition and subtraction to add and subtract rational numbers; represent addition and subtraction on a horizontal or vertical number line diagram. |
| 7.NS.A.2 | Apply and extend previous understandings of multiplication and division and of fractions to multiply and divide rational numbers. |
| 7.NS.A.3-fraction | Solve real-world and mathematical problems involving the four operations with rational numbers. |
| 7.NS.A.3-decimal | Solve real-world and mathematical problems involving the four operations with rational numbers. |

Table 12: CC Standards for Grade 7

| Standard ID | Description |
|---|---|
| 8.EE.A.2 | Use square root and cube root symbols to represent solutions to equations of the form $x^2 = p$ and $x^3 = p$, where p is a positive rational number. Evaluate square roots of small perfect squares and cube roots of small perfect cubes. Know that the square root of 2 is irrational. |
| 8.EE.C.7 | Solve linear equations in one variable. |
| 8.EE.C.8 | Analyze and solve pairs of simultaneous linear equations. |

Table 13: CC Standards for Grade 8

| Model | GPT4-generated MathCAMPS accuracy | Claude-generated MathCAMPS accuracy |
|---|---|---|
| GPT-4o | 0.910 | 0.954 |
| Claude 3 Opus | 0.887 | 0.909 |

Table 14: Performance of GPT-4o and Claude 3 Opus on the dataset genreated using Claude

## C  DATA GENERATION PIPELINE DETAILS

### C.1  GRAMMAR

We implemented a global attribute grammar in Python, where production rules are implemented as recursive Python functions. Effectively, each CC standard has its own grammar, composed of pieces from components from the global CC grammar, as well as possibly adding unique non-terminals. Each CC standard contains the following parameters:

**Description:** The description of the CC standard.

**Short description:** A shortened description of the CC standard.

**Filters:** A list of problem filters to ensure that all problems in this standard satisfy some requirement given in the Common Core description of the standard. The ProblemLength filter makes sure that the problem is within the desired length. CheckIntermediateValues filters out any problems with intermediate values greater or lesser than max_value or min_value, respectively. The ChainsOfVariables filter eliminates any problems where variables are assigned to equal exactly another variable, and nothing else. The ContainsTen filter checks if the math word problem contains numbers adding up to 10, or contains a 10 in the problem (for standards K.OA.A.4 and K.NBT.A.1, respectively).

**Transforms:** List of problem transformations applied to all symbolic structures from this standard. The NoUselessVariables transform performs dead code elimination — it removes any variables that do not contribute to the final answer by applying a simple graph reachability algorithm on a dependency graph between statements, removing statements that the answer does not depend on. The Simplify transform essentially inlines variables that are used only once.

**Expressions:** Lists non-terminals available to generate expressions in symbolic structures for this standard. For example, this can make specific binary operations (e.g. addition, division) available on that particular standard.

**Min/max value:** Specifies bounds on values for both the final answer and all intermediate values in the solution.

**Min/max number:** Specifies bounds on numeric constants sampled in the symbolic structure.

**Max depth:** Sets a maximum depth for expressions in the symbolic structure.

**Samples:** We include 2+ hand-written, standard-relevant examples of a symbolic problem followed by a relevant natural language problem generation, which we use as few-shot prompts during problem generation. We also use these prompts, but in reverse (natural language followed by symbolic problem), when we prompt GPT-4 during cycle consistency.

| | Faithful problem | Unfaithful problem |
|---|---|---|
| Cycle-consistent | 208 | 5 |
| Not cycle-consistent | 7 | 25 |

Table 15: Efficacy of Cycle Consistency

## C.2 ANSWER GRADING DURING EVALUATION

Given a solution in natural language, we first use a rule-based answer extractor to extract any model's numerical answer. In cases where a language model doesn't answer in the required format, or answers in an unexpected format, the answer is initially marked as incorrect. For all problems with incorrect answers, we use Llama-3 70B to re-extract the final answer. We few-shot prompt it with hand-generated examples of solutions and extracted final answers, and ask it to extract the final answer from the new solution. If a problem that was previously incorrect is marked as correct (given the newly extracted answer), we rerun the model on any followups the problem might have. Note that this "regrading" step can only improve accuracy from the base result, since we only run it on problems that failed under the rule-based evaluation. In practice, we found this process to have negligible false-positive rate — only in a handful of cases across all models we observed either answer extraction processes extracting the correct answer out of a wrong response (e.g., if the answer to a problem is 2, and the model responds "On day 2, Sally bought 9 dolls", the rule-based parser extracts 2 as being the model's answer, though the sentence implies its answer to be 9). On the other hand, the LLaMA-3 70B extractor greatly reduces our false negative rate in a handful of models (especially DeepSeek) which are more likely to respond in a format different from what our prompt asks for.

## C.3 COST ESTIMATE

All problems in MathCAMPS were generated using OpenAI `gpt-4-0613`, in May 2024. We estimate an approximate cost of 330 USD to generate 9607 problems (including main problems and follow-ups). This includes the cost to perform cycle consistency, and problems that are discarded by cycle consistency. This gives an average cost of 0.034 USD (3.4 cents) per cycle-consistent problem or follow-up question.

## D CYCLE CONSISTENCY EFFICACY AND FAILURE CASES

As we note in the paper, we manually evaluated 245 random problems generated by sampling a symbolic structure and then a word problem from GPT-4. The results are shown in Table 15.

### D.1 UNFAITHFUL BUT CYCLE-CONSISTENT PROBLEMS

We noted that of the five problems that were unfaithful but cycle-consistent, two had ambiguous language that led to the generation of a cycle-consistent symbolic structure. The three other problems were all from the same CC standard (4.NBT.B.6, a skill regarding the calculation of quotients and remainders).

### D.2 UNFAITHFUL AND NON-CYCLE CONSISTENT PROBLEMS

Below, we provide threeexamples of unfaithful problem generations that were unfaithful and correctly discarded by the cycle-consistency process.

**Original symbolic structure:** [[var f = (5 - 4)]][[var t = (4 - f)]][[question d = ['t']]] theme: Rocket

**Generated word problem:** In the space exploration research center, Diego has 5 rockets. He launched 1 rocket for testing purposes. After this, he orders 4 new rockets, but doesn't count the one that he just launched. How many rockets does Diego have now?

**New symbolic structure:** [[var initial_rockets = 5]][[var launched_rockets = 1]][[var ordered_rockets = 4]][[var total_rockets = (initial_rockets - launched_rockets + ordered_rockets)]][[question result = ['total_rockets']]]

**Original symbolic structure:** [[var v = (79.24 * 37.6)]][[question s = ['v']]] theme: Treasure chest

**Generated word problem:** A pirate finds a treasure chest full of golden coins. Each golden coin weighs 79.24 grams. If the total weight of the coins is 37.6 kilograms, how many golden coins are there in the treasure chest?

**New symbolic structure:** [[var weightInGrams = (37.6 * 1000)]][[var n = (weightInGrams / 79.24)]][[question numCoins = ['n']]]

**Original symbolic structure:** [[var j = (4450 / 89)]][[var c = ((69 * j) + j)]][[question b = ['c']]] theme: Sandwich

**Generated word problem:** The Sandwich Factory makes sandwiches at a rate of 4450 sandwiches every 89 days. If the factory not only produces 69 sandwiches each day, but also an additional batch of the exact same rate each day, how many sandwiches are produced in total each day?

**New symbolic structure:** [[var a = 4450]][[var b = 89]][[var c = 69]][[var d = a / b]][[var e = c + d]][[question f = ['e']]]

### D.3 FAITHFUL BUT NON-CYCLE-CONSISTENT PROBLEMS

The 7 faithful problems were discarded were because one of two issues. 6 our of 7 problems were discarded because their back-translated symbolic structure had a small structural error (e.g. and extra square bracket at the end of a variable declaration). The 7th problem had a genuine error in its back translation, which caused the new symbolic structure to have a different final answer than the original symbolic structure, causing us to discard the problem.

## E CORRELATION BETWEEN MATHCAMPS AND GSM8K

Figure 3 shows accuracies of several models on both GSM8k and MathCAMPS, along with the line of best fit. There is a strong correlation between overall accuracy in both datasets ($\rho = 0.91$, $p < 10^{-6}$), though MathCAMPS allows for many fine-grained analysis besides overall performance.

## F COMPLETE TABLES

Table 16 shows the full table from which Table 2 was extracted.

Table18 shows the full table from which Table 4 was extracted.

## G FOLLOWUP ANALYSIS

Table 17 lists model accuracies when only looking at the main problems (Main Acc.), their accuracies when only looking at the incremental followups (IFUP Acc.), their accuracies when only looking at the counterfactual followups (CFUP Acc.), and finally, the total number of followups seen by each model. The total number of followups a model sees relies on whether or not they get the main question for that followup correct. If a model does not correctly solve the main question, it is not prompted with follow-ups. Note that each followup serves as a followup to the main question, as opposed to a followup to each other.

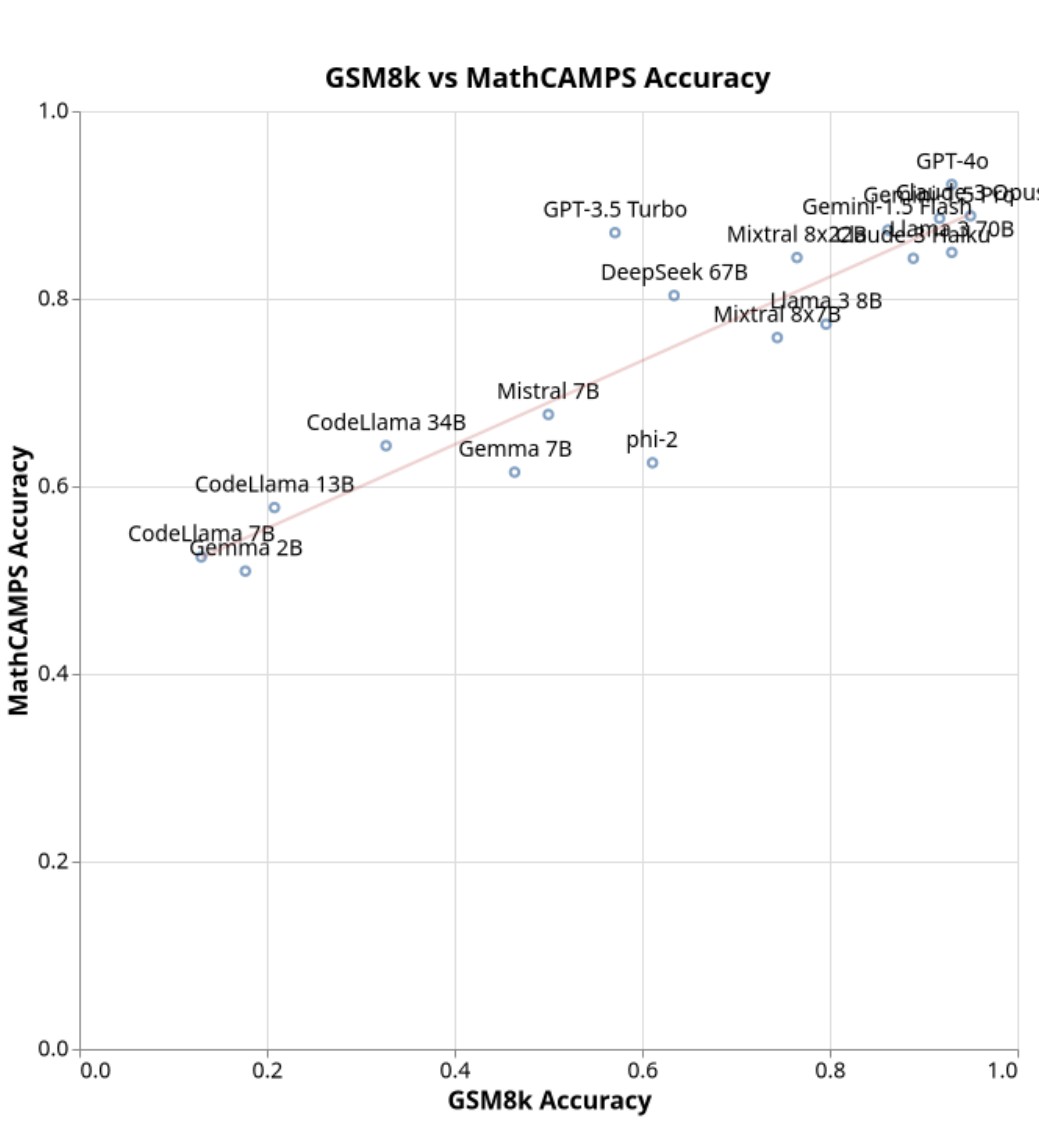

Figure 3: Relation between accuracy on GSM8k and on MathCAMPS.

Table 16: Largest model rank changes when focusing on one CC standard, in contrast to only overall performance. This is a complete version of Table 2, which only shows some models for brevity.

| Model | Top outlier skill | Rank change |
|---|---|---|
| GPT-4o | 8.EE.C.8 - Solve two-variable systems | $(1^{st} \searrow 22^{th})$ |
| Claude-3 Opus | 2.MD.B.5 - Add/sub within 100 | $(2^{nd} \searrow 18^{th})$ |
| Gemini-1.5 Pro | K.OA.A.4 - Adding to equal 10 | $(4^{th} \searrow 23^{th})$ |
| Gemini-1.5 Flash | 4.OA.B.4 - Factor pairs within 100 | $(5^{th} \searrow 26^{th})$ |
| GPT-3.5 Turbo | 6.EE.A.1 - Evaluate exponents | $(6^{th} \searrow 27^{th})$ |
| Claude-3 Sonnet | 5.NF.B.4 - Mult fractions | $(7^{th} \searrow 16^{th})$ |
| Claude-3 Haiku | 6.EE.A.1 - Evaluate exponents | $(10^{th} \searrow 20^{th})$ |
| Qwen2-Math 72B | 8.EE.C.8 - Solve two-variable systems | $(3^{rd} \searrow 26^{th})$ |
| Llama 3 70B | 3.OA.A.3 - Mul/div within 100 | $(8^{th} \searrow 21^{th})$ |
| Mixtral 8x22B | 8.EE.C.8 - Solve two-variable systems | $(9^{th} \searrow 21^{th})$ |
| Qwen2-Math 7B | 8.EE.C.8 - Solve two-variable systems | $(11^{th} \searrow 25^{th})$ |
| DeepSeek 67B | K.NBT.A.1 - Decompose into 10s | $(12^{th} \nearrow 1^{st})$ |
| DeepSeek Math 7B Base | 8.EE.C.8 - Solve two-variable systems | $(13^{th} \searrow 28^{th})$ |
| NuminaMath 7B TIR | 8.EE.C.8 - Solve two-variable systems | $(14^{th} \searrow 27^{th})$ |
| Llama 3 8B | K.OA.A.4 - Adding to equal 10 | $(15^{th} \nearrow 3^{rd})$ |
| Mixtral 8x7B | 6.EE.A.1 - Evaluate exponents | $(16^{th} \searrow 26^{th})$ |
| InternLM-Math Base 20B | 2.NBT.B.5 - Add/sub within 100 | $(17^{th} \nearrow 2^{nd})$ |
| Llemma 34B | 3.OA.A.3 - Mul/div within 100 | $(18^{th} \nearrow 1^{st})$ |
| Mistral 7B | 1.OA.A.1 - Add/sub within 20 | $(19^{th} \searrow 26^{th})$ |
| DeepSeek Coder 33B | 6.EE.A.1 - Evaluate exponents | $(20^{th} \nearrow 3^{rd})$ |
| CodeLlama 34B | 6.EE.A.1 - Evaluate exponents | $(21^{th} \nearrow 11^{th})$ |
| phi-2 | K.OA.A.4 - Adding to equal 10 | $(22^{th} \nearrow 4^{th})$ |
| Llemma 7B | 6.EE.A.1 - Evaluate exponents | $(23^{th} \nearrow 5^{th})$ |
| Gemma 7B | K.OA.A.5 - Add/sub within 5 | $(24^{th} \nearrow 6^{th})$ |
| CodeLlama 13B | 1.OA.A.2 - Add three nums within 20 | $(25^{th} \nearrow 14^{th})$ |
| InternLM-Math Base 7B | 4.OA.B.4 - Factor pairs within 100 | $(26^{th} \nearrow 15^{th})$ |
| CodeLlama 7B | 8.EE.C.8 - Solve two-variable systems | $(27^{th} \nearrow 15^{th})$ |
| Gemma 2B | 8.EE.C.8 - Solve two-variable systems | $(28^{th} \nearrow 11^{th})$ |

| Vendor | Model | Main Acc. | IFUP Acc. | CFUP Acc. | Total FUPs seen |
|--------|-------|-----------|-----------|-----------|-----------------|
| Anthropic | Claude-3 Opus | 0.89 | 0.90 | 0.88 | 4142 |
| Anthropic | Claude-3 Sonnet | 0.86 | 0.86 | 0.87 | 3964 |
| Anthropic | Claude-3 Haiku | 0.84 | 0.88 | 0.87 | 3819 |
| DeepSeek | DeepSeek Coder 33B | 0.65 | 0.79 | 0.85 | 1022 |
| DeepSeek | DeepSeek 67B | 0.80 | 0.87 | 0.88 | 3286 |
| EleutherAI | Llemma 7B | 0.62 | 0.67 | 0.79 | 2835 |
| EleutherAI | Llemma 34B | 0.71 | 0.79 | 0.85 | 3229 |
| Google | Gemini-1.5 Pro | 0.89 | 0.91 | 0.89 | 4140 |
| Google | Gemini-1.5 Flash | 0.87 | 0.89 | 0.87 | 4083 |
| Google | Gemma 2B | 0.51 | 0.29 | 0.54 | 2044 |
| Google | Gemma 7B | 0.62 | 0.55 | 0.60 | 2786 |
| Meta | Llama 3 8B | 0.77 | 0.84 | 0.80 | 3476 |
| Meta | Llama 3 70B | 0.85 | 0.87 | 0.84 | 3939 |
| Meta | CodeLlama 7B | 0.52 | 0.69 | 0.86 | 617 |
| Meta | CodeLlama 13B | 0.58 | 0.75 | 0.80 | 2451 |
| Meta | CodeLlama 34B | 0.64 | 0.82 | 0.88 | 844 |
| Microsoft | phi-2 | 0.63 | 0.48 | 0.78 | 2873 |
| Mistral | Mistral 7B | 0.68 | 0.72 | 0.80 | 3090 |
| Mistral | Mixtral 8x7B | 0.76 | 0.80 | 0.82 | 3439 |
| Mistral | Mixtral 8x22B | 0.84 | 0.86 | 0.83 | 3948 |
| OpenAI | GPT-4o | 0.92 | 0.92 | 0.90 | 4358 |
| OpenAI | GPT-3.5 Turbo | 0.87 | 0.85 | 0.86 | 4063 |
| InternLM | InternLM-Math Base 7B | 0.58 | 0.67 | 0.84 | 2628 |
| InternLM | InternLM-Math Base 20B | 0.74 | 0.78 | 0.86 | 3409 |
| Qwen | Qwen2-Math 7B | 0.83 | 0.88 | 0.89 | 3774 |
| Qwen | Qwen2-Math 72B | 0.89 | 0.90 | 0.91 | 4119 |
| Numina | NuminaMath 7B TIR | 0.78 | 0.82 | 0.86 | 3593 |
| DeepSeek | DeepSeek Math 7B Base | 0.78 | 0.84 | 0.88 | 3583 |

Table 17: Model performance on our mathematical dialogue task, where the model must answer follow-up questions besides the initial problem. CFUP and IFUP Acc. indicate the accuracy of the model on counterfactual and incremental followups respectively. Total FUPs refers to the total number of follow up questions each model sees, which differs by model since a model, only sees a followup question if it answers the main question correctly.

Table 18: Model performance on our mathematical dialogue task, where the model must answer follow-up questions besides the initial problem. The Table is a complete version of Table 4, which only shows some models for brevity.

| Model | Acc. with follow-ups | Largest accuracy drop w/ follow-ups | |
|---|---|---|---|
| GPT-4o | 0.82 | 5.NF.A.1 - Add/sub fractions | 0.86 ↘0.58) |
| Claude-3 Opus | 0.76 | 7.NS.A.1-fraction - Add/sub with fractions | 0.54 ↘0.23) |
| Gemini-1.5 Pro | 0.77 | 5.OA.A.1 - Evaluating with parentheses | 0.95 ↘0.69) |
| Gemini-1.5 Flash | 0.76 | 7.NS.A.1-fraction - Add/sub with fractions | 0.74 ↘0.37) |
| GPT-3.5 Turbo | 0.71 | 7.NS.A.1-fraction - Add/sub with fractions | 0.70 ↘0.21) |
| Claude-3 Sonnet | 0.72 | 7.NS.A.1-fraction - Add/sub with fractions | 0.44 ↘0.10) |
| Claude-3 Haiku | 0.70 | 7.NS.A.2 - Mult/div with fractions | 0.55 ↘0.26) |
| Qwen2-Math 72B | 0.78 | 5.NF.A.1 - Add/sub fractions | 0.49 ↘0.23) |
| Llama 3 70B | 0.69 | 4.NF.A.2 - Compare two fractions | 0.99 ↘0.66) |
| Mixtral 8x22B | 0.69 | 7.NS.A.1-fraction - Add/sub with fractions | 0.69 ↘0.17) |
| Qwen2-Math 7B | 0.71 | 5.NF.A.2 - Add/sub fraction word problems | 0.41 ↘0.17) |
| DeepSeek 67B | 0.68 | 6.NS.B.3 - Add/sub/mult/div decimals | 0.59 ↘0.37) |
| DeepSeek Math 7B Base | 0.65 | 5.NF.B.4 - Mult fractions | 0.81 ↘0.57) |
| NuminaMath 7B TIR | 0.62 | 5.NF.A.2 - Add/sub fraction word problems | 0.44 ↘0.18) |
| Llama 3 8B | 0.58 | 4.NF.A.2 - Compare two fractions | 0.90 ↘0.52) |
| Mixtral 8x7B | 0.58 | 7.NS.A.2 - Mult/div with fractions | 0.60 ↘0.28) |
| InternLM-Math Base 20B | 0.58 | 7.NS.A.2 - Mult/div with fractions | 0.59 ↘0.26) |
| Llemma 34B | 0.55 | 5.NF.B.4 - Mult fractions | 0.68 ↘0.31) |
| Mistral 7B | 0.48 | 7.NS.A.1-decimal - Add/sub with decimals | 0.91 ↘0.50) |
| DeepSeek Coder 33B | 0.60 | 3.OA.A.3 - Mul/div within 100 | 0.95 ↘0.81) |
| CodeLlama 34B | 0.60 | 5.NF.B.4 - Mult fractions | 0.51 ↘0.39) |
| phi-2 | 0.39 | 3.NBT.A.2 - Add/sub within 1000 | 0.71 ↘0.23) |
| Llemma 7B | 0.42 | 5.NF.B.4 - Mult fractions | 0.58 ↘0.21) |
| Gemma 7B | 0.33 | 7.NS.A.1-decimal - Add/sub with decimals | 0.91 ↘0.32) |
| CodeLlama 13B | 0.43 | 4.NBT.B.4 - Add/sub multi-digit nums | 0.81 ↘0.49) |
| InternLM-Math Base 7B | 0.42 | 7.NS.A.1-decimal - Add/sub with decimals | 0.82 ↘0.47) |
| CodeLlama 7B | 0.49 | 2.NBT.B.7 - Add/sub within 100 | 0.80 ↘0.67) |
| Gemma 2B | 0.24 | 3.NBT.A.2 - Add/sub within 1000 | 0.93 ↘0.26) |

| Model Name | Citation |
| --- | --- |
| GPT-4o | (Achiam et al., 2023) |
| Claude-3 Opus | (Anthropic, 2024) |
| Gemini-1.5 Pro | (Team et al., 2023) |
| Gemini-1.5 Flash | (Team et al., 2023) |
| GPT-3.5 Turbo | (Achiam et al., 2023) |
| Claude-3 Sonnet | (Anthropic, 2024) |
| Claude-3 Haiku | (Anthropic, 2024) |
| Qwen2-Math 72B | (Yang et al., 2024) |
| Llama 3 70B | (Touvron et al., 2023) |
| Mixtral 8x22B | (Jiang et al., 2024) |
| Qwen2-Math 7B | (Yang et al., 2024) |
| DeepSeek 67B | (Bi et al., 2024) |
| DeepSeek Math 7B Base | (Shao et al., 2024) |
| NuminaMath 7B TIR | (Fleureau et al., 2024) |
| Llama 3 8B | (Touvron et al., 2023) |
| Mixtral 8x7B | (Jiang et al., 2024) |
| InternLM-Math Base 20B | (Ying et al., 2024) |
| Llemma 34B | (Azerbayev et al., 2023) |
| Mistral 7B | (Jiang et al., 2023) |
| DeepSeek Coder 33B | (Guo et al., 2024) |
| CodeLlama 34B | (Roziere et al., 2023) |
| phi-2 | (Li et al., 2023) |
| Llemma 7B | (Azerbayev et al., 2023) |
| Gemma 7B | (Team et al., 2024) |
| CodeLlama 13B | (Roziere et al., 2023) |
| InternLM-Math Base 7B | (Ying et al., 2024) |
| CodeLlama 7B | (Roziere et al., 2023) |
| Gemma 2B | (Team et al., 2024) |

Table 19: Model names and their citations.

