# OpenReview forum: "MathCAMPS: Fine-grained Synthesis of Mathematical Problems From Human Curricula"
_ICLR.cc/2025/Conference — Submitted to ICLR 2025_

### Official Review · Reviewer_abf6 · 2024-11-04

**Soundness:** 3
**Presentation:** 3
**Contribution:** 3
**Rating:** 6
**Confidence:** 4

**Summary:**

This paper introduces a scalable approach for generating high-quality synthetic mathematical problems. The process begins by using a math standard to develop a formal grammar, which is used to sample symbolic problems with answers. These symbolic problems are then transformed into word problems using large language models (LLMs). A cycle-consistency method validates the faithfulness of the generated problems. Additionally, follow-up questions derived from symbolic structures are transformed into word problems, creating a new type of mathematical dialogue task to assess understanding robustness. Extensive evaluation shows that even advanced models struggle with follow-up questions. The paper also analyzes the development of specific mathematical skills by using different training checkpoints of the Pythia 12B model.

**Strengths:**

- MathCAMPS Benchmark: The proposed MathCAMPS benchmark is precisely aligned with the K-8 Mathematics Common Core (CC) Standards, enabling a comprehensive evaluation of mathematical skills. This structure supports a detailed analysis of proficiency exhibited by Large Language Models (LLMs) across specific mathematical reasoning capabilities.

- Grammar-Based Skill Encoding: By encoding CC skills in a structured grammar (symbolic representation), MathCAMPS can generate a wide array of targeted problems that address specific skills, such as decimal addition or solving equations with fractions.

- Cycle-Consistency Validation: MathCAMPS employs a cycle-consistency method to confirm that word problems accurately represent their original symbolic counterparts. By prompting the language model to translate a word problem back into symbolic form and then comparing results, this method effectively reduces misrepresentation and enhances problem fidelity.
- Mathematical Dialogue for Enhanced Understanding: MathCAMPS introduces two types of “mathematical dialogue” tasks to probe deeper understanding.
   -- Counterfactual Questions: These questions modify aspects of the original problem, testing the model's adaptability to new conditions.
Incremental Questions: These questions add information to the problem, requiring the model to integrate prior context with new details.
   -- Incremental Questions: These questions add information to the problem, requiring the model to integrate prior context with new details.

However, apart from using CC standards and dialog setting, the novelty of the work is low.

**Weaknesses:**

This work has gone in a similar direction as many papers, where researchers have looked back at the progress of math reasoning and found that substantial progress has not been achieved as claimed by tall IMO claims.  For example recently [1] found a way to synthetically generate symbolic math dataset with controlled perturbations, train and test some BERT based models and also benchmark SOTA LLMs.  Previously the type of inconsistency was done  by [3] where authors showed that vanilla transformers can not multiply numbers while [2] showed success in solving integrals.
While these papers are mostly symbolic problems without much of natural language, other work, such as GSM-Symbolic [3], MORE [4] has provided a vast coverage of different variations that probes the robustness of arithmetic reasoning abilities. MORE provides a vast ontology of perturbations and GSM-symbolic provides many useful templates. Also, while GSM-symbolic is fairly recent, MORE has been published many months back. Given the similarity, it should be mentioned and compared with.

Unfortunately, the paper therefore is not as novel as it claims to be. Other than utilizing CC standards and the dialog setting, contributions are questionable.
More questions:
1. How faithful is the representation from CC NL guidelines to the formal grammar? Have the authors evaluated this step independently?
2. LLM generations are evaluated/flitered using cycle consistency. How did you filter our hallucinations, other errors? Again, what is the comparison with MORE?

The insights from training Pythia feels much more interesting than the other parts of the paper, though it consitutes a small body of the entire work.


[1] A Symbolic Framework for Evaluating Mathematical Reasoning and Generalisation with Transformers

[2] Deep Learning for Symbolic Math
[3]	Analyzing the Nuances of Transformers' Polynomial Simplification Abilities
[4] GSM-Symbolic: Understanding the Limitations of Mathematical Reasoning in Large Language Models
[5] Evaluating LLMs' Mathematical and Coding Competency through Ontology-guided Interventions

**Questions:**

See above.

---

> ### Author Response · Authors · 2024-11-20
>
> We thank the reviewer for the thorough evaluation of our work! We're pleased that you found our use of the Common Core standards and the dialogue setting novel, and appreciated the insights from the Pythia training. We respond to concerns and questions below.
>
> > This work has gone in a similar direction as many papers, [...] GSM-Symbolic [3], MORE [4] has provided a vast coverage of different variations that probes the robustness of arithmetic reasoning abilities. MORE provides a vast ontology of perturbations and GSM-symbolic provides many useful templates. Also, while GSM-symbolic is fairly recent, MORE has been published many months back. Given the similarity, it should be mentioned and compared with.
>
> We thank the reviewer for pointing out these pieces of related work. We included a discussion of GSM-Symbolic and GSMore in our Related Work section (see Section 2, paragraph "LLM-generated synthetic datasets for LLMs" in the updated paper). For convenience, we also include the discussion here. Both GSM-Symbolic (only released after the ICLR deadline) and GSMore are interesting works that focus on probing the robustness of LLMs by evaluating how their answers change under different semantically-preserving transformations of existing problems. Instead of relying on existing datasets, we note that MathCAMPS generates problems without the need for a seed dataset. Moreover, our focus is on providing an evaluation that is grounded on a human curriculum, which is not a desiderata of either of these works. This is our focus, more than robustness per se, although our follow-up question analysis also shows failures in robustness in most LLMs. We strongly believe that these are complementary analyses of mathematical skills of LLMs.
>
> > Unfortunately, the paper therefore is not as novel as it claims to be.
>
> We are happy to revise our claims if the reviewer believes we stated more novelty than is warranted. However, we refer to our stated contributions at the end of the introduction: (1) a synthetic evaluation that is fully grounded on a human curriculum, (2) the cycle consistency method for evaluating faithfulness, providing a fully synthetic pipeline (note that GSMore required manual intervention to filter out errors in generation, showing the challenge of ensuring data quality in a fully automated pipeline. We tackle this challenge here, allowing our pipeline to scale up cheaply), (3) a method to generate follow-up questions, and (4) the analyses of mathematical skills emerging in Pythia checkpoints. We believe these are fair claims in light of where existing work is at.
>
> > How faithful is the representation from CC NL guidelines to the formal grammar? Have the authors evaluated this step independently?
>
> When encoding the standards, we made sure to look at teacher-created worksheets online (several are available on a CC standard basis on websites like https://www.commoncoresheets.com/), to make sure that our problems reflected the content from real examples. Evaluating this step would likely involve a large-scale study with teachers. Having the perspectives of teachers would be especially important for potential educational applications of our work. For instance, as we mention in the paper, we believe there is a potential to use our pipeline to generate custom problems for human students, fixing the mathematical skill (which can follow their curriculum) but varying the story in a customized way. This application of synthetic problems for evaluating or teaching human students is, however, a separate endeavor that is worth doing right — we would need to recruit an expert population of teachers in order to annotate a spectrum of different problems, or recruit students of appropriate grade levels in order to establish psychometric validity of the problems, if they were to complement existing worksheets. While these evaluations would go beyond the scope of the current work, we will expand on our discussion into what these would entail for future work in our conclusion.

---

> > ### Author Response · Authors · 2024-11-20
> >
> > > LLM generations are evaluated/flitered using cycle consistency. How did you filter our hallucinations, other errors? Again, what is the comparison with MORE?
> >
> > We found that our cycle consistency method was simple and effective against a large number of failure modes of synthetic generation. To ground the discussion here, say we start with a problem p; generate word problem w using an LLM. In cycle-consistency, we generate a symbolic problem p' only from w, and check if answer(p) = answer(p'). For hallucinations (e.g., w has incorrect information, or the LLM made up facts not present in p problem), the probability of passing cycle consistency is extremely low, because in generating p' the LLM would need to exactly undo the same hallucinations without having access to the original problem p. The same happens with ambiguous problems: if w admits multiple interpretations, or misses information that was present in p, the LLM cannot reconstruct a problem equivalent to p in the cycle consistency stage (it would have to make a guess without seeing the original problem).
> >
> > As we show in our human evaluation of data quality (Section 3.2), this process was highly effective: only 2.3% of the problems after cycle consistency are unfaithful to the original symbolic problems. This is a smaller rate than many human-created datasets, especially crowdsourced ones.
> >
> > In contrast, MORE relied on a human verification to ensure similar high quality (see Section 4.2 of their paper). After the automated phase of their pipeline (with generation + GPT-4 checking for errors), only 33% of the problems were fully correct, with the remaining needing discarding or editing by humans. As the authors point out, this makes their pipeline semi-automated, in contrast to MathCAMPS which is fully automated.
> >
> > Thank you again, and we're happy to engage further in the discussion if there's anything else we can clarify.

---

> > > ### Comment · Reviewer_abf6 · 2024-11-25
> > >
> > > > To ground the discussion here, say we start with a problem p; generate word problem w using an LLM. In cycle-consistency, we generate a symbolic problem p' only from w, and check if answer(p) = answer(p'). For hallucinations (e.g., w has incorrect information, or the LLM made up facts not present in p problem), the probability of passing cycle consistency is extremely low, because in generating p' the LLM would need to exactly undo the same hallucinations without having access to the original problem p.
> > >
> > > Sorry I did not understand what do you mean here fully. The probability of coming up with an answer using a wrong path can not ve extremely low. In general, it is better to check the path or CoT as well. Can you explain why you say the probability passing cycle consistency is low?

---

> > > > ### Author Response · Authors · 2024-11-25
> > > >
> > > > We sincerely thank the reviewer for considering our response! We hope to clarify the remaining comments below.
> > > >
> > > > > Sorry I did not understand what do you mean here fully. The probability of coming up with an answer using a wrong path can not ve extremely low. In general, it is better to check the path or CoT as well. Can you explain why you say the probability passing cycle consistency is low?
> > > >
> > > > We believe there are two points for clarification here:
> > > >
> > > > * First, note that our pipeline for generating the dataset *nowhere relies on an LLM to produce CoT traces or even final answers* to the problems. To solve the problems, we rely on a trusted, deterministic, symbolic solver (SymPy for most cases). Thus, we sample symbolic problems from our grammar, and obtain their final answers in a reliable way that does not depend on an LLM.
> > > > * From the symbolic problem, we then use an LLM to translate that into a word problem (now in natural language). Doing this naïvely incurs a risk of the word problem not faithfully representing the symbolic problem. Here is an example to illustrate this issue:
> > > >
> > > > Symbolic problem (sampled from a grammar);
> > > > ```
> > > > [[var x = 10]]
> > > > [[var y = x + 5]]
> > > > [[question y]]
> > > > theme: Candy
> > > > ```
> > > > Answer: 15 (obtained symbolically)
> > > >
> > > > Candidate word problem (sampled from an LLM):
> > > > ```
> > > > John received 10 pieces of chocolate from his neighbor on Halloween. He then received 15 other pieces of candy from his mom. How many pieces of candy did he end up with in total?
> > > > ```
> > > >
> > > > Note that this problem does not correctly represent the original symbolic problem. Thus, its answer is not 15 (it would be 25). This incoherence is the situation that the cycle consistency method tries to avoid. In cycle consistency, we start with the candidate word problem above, and prompt an LLM to generate an equivalent *symbolic problem, without having access to the original symbolic problem*. We then check whether the new symbolic problem has the same answer as the initial one (again, with a deterministic symbolic solver, not an LLM). The reason why it is highly unlikely that this produces a false positive is that the LLM would have to undo the mistake done in the previous step without having any information about what that mistake might have been (since it does not see the original problem, and just makes a prediction on what that would be based on the word problem). For example, in cases where the word problem *misses information* that was present in the symbolic problem, the LLM during cycle consistency has no plausible way to reconstruct that information to make the unfaithful problem pass the consistency check. We refer again to our human evaluation of data quality (Section 3.2), which confirmed that this process was highly effective: only 2.3% of the problems after cycle consistency are unfaithful to the original symbolic problems, suggesting a higher quality than many human-created datasets.
> > > >
> > > > We thank you again for engaging with us and revising your score, and would be happy to clarify any further concerns!

---

> > > > > ### Comment · Reviewer_abf6 · 2024-11-27
> > > > > **Cycle Consistency**
> > > > >
> > > > > Thanks. That clarifies bit more about the cycle consistency.
> > > > >
> > > > > It is not a major concern though. However, can't one use the new word problem+generated symbolic problem to begin with? Assume you have a symbolic problem $sym_p$, from there a word problem is generated $word_p$, which corresponds to $sym_q$.  $sym_q$ differs from $sym_p$ in some aspects. If $word_p$ is semantically and logically equivalent to $sym_q$, why not use $\langle word_p, sym_q\rangle$ itself.
> > > > > I feel in some ways, reviewer FyCa has also questioned the possible issues about cycle-consistency.

---

> > > > > > ### Author Response · Authors · 2024-12-04
> > > > > >
> > > > > > Thanks for the reply, and we're glad we were able to clarify cycle consistency! We comment on the last suggestion below.
> > > > > >
> > > > > > > It is not a major concern though. However, can't one use the new word problem+generated symbolic problem to begin with? Assume you have a symbolic problem sym_p, from there a word problem is generated word_p, which corresponds to sym_q. sym_q differs from sym_p in some aspects. If sym_q is semantically and logically equivalent to word_p, why not use (word_p, sym_q) itself.
> > > > > >
> > > > > > This is an interesting idea! Unfortunately, when cycle consistency fails, we are also not guaranteed that word_p is a valid or sensible problem (or that sym_q is logically equivalent to it). For example, consider this example that happened in a standard relating to polygons:
> > > > > >
> > > > > > ```
> > > > > > symbolic problem:
> > > > > > [[eq 470 = (((((((39 + d) + 38) + 52) + 40) + 68) + 54) + 92)]]
> > > > > > [[question k = ['d']]]
> > > > > >
> > > > > > Concept: create a math problem involving perimeters of polygons including finding the perimeter given the side lengths, or finding an unknown side length
> > > > > >
> > > > > > Word Problem (LLM-generated): An octagon has side lengths of 39m, 38m, 52m, 40m, 68m, 54m, and 92m. What is the length of its eighth side?
> > > > > >
> > > > > > symbolic struct generated in cycle consistency:
> > > > > > [[eq 383 = 39 + 38 + 52 + 40 + 68 + 54 + 92 + a]]
> > > > > > [[question h = ['a']]]
> > > > > > ```
> > > > > >
> > > > > > Note that the word problem here has a crucial issue: it fails to mention the total perimeter of the octagon (which would be 470 according to the symbolic structure). Without that information, the problem is unsolvable: we cannot determine the last side. Thus, when the LLM tries to generate a symbolic structure for cycle consistency, it makes up some other constant, 383, to be the total length. Note that 39 + 38 + 52 + 40 + 68 + 54 + 92 = 383, so the solution to the new symbolic problem is `a = 0`. This fails cycle consistency. But also note that word_p and sym_q are also not a suitable pair, since the word problem doesn't really have a solution.
> > > > > >
> > > > > > We hope this illustrates why we discard problems that fail cycle consistency, as opposed to using them along with the new symbolic structure.
> > > > > >
> > > > > > > I feel in some ways, reviewer FyCa has also questioned the possible issues about cycle-consistency.
> > > > > >
> > > > > > We note that the example above also illustrates why the possibility raised by reviewer FyCa, that a strong model might ignore and undo the errors during cycle consistency, does not apply: since the model cannot see the original problem (where the perimeter was 470), it simply does not have enough information during cycle consistency to reconstruct the original problem. This is regardless of how strong the model is.
> > > > > >
> > > > > > We hope this clarifies these last points. Thanks again for the engagement!

---

> > ### Comment · Reviewer_abf6 · 2024-11-25
> > **Thank you**
> >
> > Thank you for your response. This clarifies a few aspects. I am increasing the score for now. But, I am not sure how much really this adds to the current community knowledge. How futuristic the benchmark overall is.

---

### Official Review · Reviewer_47B3 · 2024-11-04

**Soundness:** 3
**Presentation:** 3
**Contribution:** 3
**Rating:** 6
**Confidence:** 3

**Summary:**

This paper presents MathCAMPS, a framework for the fine-grained synthesis of mathematical problems based on human curricula. It employs a symbolic-based approach to generate scalable evaluation data for models. The authors introduce a cycle-consistency method to validate the faithfulness of the generated problems. Additionally, they conduct a series of experiments to validate existing models on the proposed benchmark.

**Strengths:**

1. The motivations of this paper are well-founded: 1. Preventing data contamination 2. Providing fine-grained evaluation.
That's exactly the problem that cannot handled by existing mathematical benchmarks.

2. The pipeline proposed by the authors effectively addresses the identified challenges. In particular, the cycle-consistency is novel and solid, significantly enhancing the quality of the generated data.

3. Furthermore, the experimental section of the paper is comprehensive and offers valuable insights.

**Weaknesses:**

1. While the proposed method demonstrates a degree of scalability, it is primarily limited to K-8 level mathematics. The experiments conducted in this study show a strong consistency with evaluations from the GSM8K benchmark. However, I don't think as more powerful models, such as OpenAI's o1, emerge, this benchmark will present significant challenges. Therefore, the potential for extending this framework to more complex problems is a key area for improvement in this work.

2. Additionally, the paper's heavy reliance on the Common Core (CC) standards results in an inability to generate mathematical problems outside of these guidelines. Consequently, more complex problems (those above K-8) or those that fall within K-8 but are not included in the CC standards cannot be synthesized.

**Questions:**

1. How does the author ensure the correctness of encoding the CC Standard into rules? Did the author conduct manual cross-validation or sampling tests?

2. Are there any methods to extend this pipeline to more difficult topics?

---

> ### Author Response · Authors · 2024-11-20
>
> We thank the reviewer for the positive remarks on our work, and for the detailed comments! We’ve addressed the points for clarification below:
>
> > While the proposed method demonstrates a degree of scalability, it is primarily limited to K-8 level mathematics. The experiments conducted in this study show a strong consistency with evaluations from the GSM8K benchmark. However, I don't think as more powerful models, such as OpenAI's o1, emerge, this benchmark will present significant challenges. Therefore, the potential for extending this framework to more complex problems is a key area for improvement in this work.
>
> We agree with the reviewer that frontier models overall perform well on MathCAMPS. However, the results still show large variability in performance when looking at individual skill, demonstrating the importance of fine-grained benchmarks to shed light on specific capabilities and failure modes. Specifically, the granularity from the evaluations in MathCAMPS allows for interpretability analysis in LLM math reasoning skills which isn’t frequently conducted. Moreover, as our analysis with Pythia shows, our dataset provides a unique opportunity to understand training dynamics, and how particular skills are learned, which is interesting even for simple mathematical skills.
>
> > Additionally, the paper's heavy reliance on the Common Core (CC) standards results in an inability to generate mathematical problems outside of these guidelines. Consequently, more complex problems (those above K-8) or those that fall within K-8 but are not included in the CC standards cannot be synthesized.
>
> The MathCAMPS pipeline can be expanded to problems outside of K-8. The Common Core covers grades K-12, and other curricula exist for subjects like Calculus and Linear Algebra that would enable the creation of grammars and solvers to support MathCAMPS-Calculus and MathCAMPS-LinearAlgebra versions. The main challenge would be standards within the Common Core that are more open-ended (e.g. interpreting tables), since conceptual questions are difficult to grade.
>
> > How does the author ensure the correctness of encoding the CC Standard into rules? Did the author conduct manual cross-validation or sampling tests?
>
> Thanks for the good question! When encoding the standards, we made sure to look at teacher-created worksheets online (several are available on a CC standard basis on websites like https://www.commoncoresheets.com/), to ensure that our problems reflected the content from real examples. Evaluating this step would likely involve a large-scale study with teachers. Having the perspectives of teachers would be especially important for potential educational applications of our work. For instance, as we mention in the paper, we believe there is a potential to use our pipeline to generate custom problems for human students, fixing the mathematical skill (which can follow their curriculum) but varying the story in a customized way. This application of synthetic problems for evaluating or teaching human students is, however, a separate endeavor that is worth doing right — we would need to recruit an expert population of teachers in order to annotate a spectrum of different problems, or recruit students of appropriate grade levels in order to establish psychometric validity of the problems, if they were to complement existing worksheets. While these evaluations would go beyond the scope of the current work, we will expand on our discussion into what these would entail for future work in our conclusion.
>
> > Are there any methods to extend this pipeline to more difficult topics?
>
> Yes! The basic premise for MathCAMPS is: (1) randomly sample a symbolic problem, (2) translate the symbolic problem into a word problem, (3) back-translate the word problem into a symbolic problem, (4) check for answer agreement between the two symbolic problems. Steps 1, 2, and 4 rely on the existence of symbolic solvers for a certain subtopic in math (for example, the python pyro library for probabilistic problems). As long as we have access to a symbolic solver for a subtopic and a strong LLM, more problems can be generated. Generally, Computer Algebra Systems like sympy (the one we use) already cover a very wide range of topics (including calculus, linear algebra, geometry, physics, combinatorics, etc), and these all can thus be integrated with the same ideas.
>
> We thank the reviewer again, and we'd be happy to engage further if there are any outstanding questions or concerns!

---

### Official Review · Reviewer_rFvy · 2024-11-04

**Soundness:** 3
**Presentation:** 3
**Contribution:** 3
**Rating:** 6
**Confidence:** 4

**Summary:**

The paper "MATHCAMPS: Fine-Grained Synthesis of Mathematical Problems from Human Curricula" presents a novel approach to generating mathematical problems that align with educational standards, specifically the Common Core. The authors introduce a pipeline that synthesizes symbolic problems, generates follow-up questions, and translates these into natural language, enhancing the interaction between LLMs and mathematical reasoning tasks.

Key contributions of the paper include:

- Problem Generation Framework: The authors develop a method to create high-quality mathematical problems based on established curricula, addressing the challenge of data contamination in existing benchmarks.

- Cycle Consistency Validation: A unique cycle-consistency check is proposed to ensure the faithfulness of the generated problems, enhancing the reliability of the evaluation process.

- Follow-Up Question Integration: The study emphasizes the importance of follow-up questions in assessing the robustness of LLMs, revealing significant performance drops when models are required to handle these additional queries.

Overall, the paper contributes to the understanding of mathematical reasoning in LLMs and provides a framework for future research in generating and evaluating mathematical problems.

**Strengths:**

- This paper investigates a critical topic: how to synthesize test sets in order to reduce the risk of leakage associated with publicly available test sets in large model training.

- The MathCAMPS test set, designed based on the Mathematics Common Core Standards for K-8 grades, allows for a fine-grained assessment of models' mathematical reasoning abilities, which is highly significant.

- The designed mathematical dialogue and cycle-consistency check are sensible.

**Weaknesses:**

- The synthesized problems depend on the capabilities of GPT-4. If the questions are particularly difficult, such as those on the AIME, this framework may fail due to insufficient synthesis model capabilities. However, manually designed questions can mitigate this issue.

- MathCAMPS is only compared to GSM8K, which is insufficient, as GSM8K contains only simple elementary algebra problems.

- It cannot be guaranteed that the synthesized mathematical problems are always reasonable or that the answers will consistently match.

**Questions:**

- When sampling different questions, the model's performance is expected to show subtle variations. The authors should sample multiple test sets to assess the variance in the model's performance.

- The authors should discuss how the framework can be extended when we need to synthesize more complex problems.

---

> ### Author Response · Authors · 2024-11-20
>
> We thank the reviewer for their encouraging comments and detailed suggestions! We’ve addressed all the specific points below:
>
> > The synthesized problems depend on the capabilities of GPT-4. If the questions are particularly difficult, such as those on the AIME, this framework may fail due to insufficient synthesis model capabilities. However, manually designed questions can mitigate this issue.
>
> We note that our framework does not depend on the capability of the generator model to *solve* the problem. Rather, it only needs to be able to translate the symbolic problem into a word problem (and back, for cycle consistency). This is a much easier task, and in fact we get many problems that GPT-4o does not solve, as our analysis shows, despite it having generated the original problem. This is because we obtain the answer symbolically, rather than relying on the model itself. MathCAMPS can also be expanded to more challenging domains like calculus and linear algebra, using appropriate solvers (sympy, which we already use, can in fact solve many problems in these domains), again not relying on the model to solve them correctly.
>
> > MathCAMPS is only compared to GSM8K, which is insufficient, as GSM8K contains only simple elementary algebra problems.
>
> We chose to compare to GSM8K since our benchmark covers skills at the K-8 level, which is somewhat similar to the coverage offered by GSM8K. Benchmarks like MATH are inherently more challenging (given their content), making them an unsuitable point of comparison.
>
> > It cannot be guaranteed that the synthesized mathematical problems are always reasonable or that the answers will consistently match.
>
> While there is no formal guarantee, we manually evaluated 245 problems and found that 97.7% of the problems were high quality (i.e. the numerical answer matched the word problem). Notably, human-curated datasets often also have unreasonable questions. [1] shows a few examples of such problems found in GSM8K and SVAMP, suggesting that their noise rate can potentially be higher than what we get with MathCAMPS (2.3% false positives).
>
> [1] https://huggingface.co/datasets/Cleanlab/bad_data_gsm8k_svamp.csv
>
> > When sampling different questions, the model's performance is expected to show subtle variations. The authors should sample multiple test sets to assess the variance in the model's performance.
>
> We ran a study on this in Appendix B ("Familiarity Bias"), generating a separate dataset with a separate model. We used Claude Opus to generate a 10% scale dataset, which we evaluated Claude and GPT-4o on, and overall found our results to be robust regardless of the model used to generate the dataset.
>
> > The authors should discuss how the framework can be extended when we need to synthesize more complex problems.
>
> We address how the framework can be expanded in the conclusion. Specifically, the framework can be expanded to subtopics in math that have solver libraries available (e.g. calculus and linear algebra). This is because the underlying idea of sampling a symbolic problem, translating it into a word problem, and back translating it back to the symbolic structure is domain-agnostic.
>
> Thanks again for the thorough review! We're happy to engage further if you have any remaining concerns.

---

> > ### Comment · Reviewer_rFvy · 2024-11-27
> > **Official Comment by Reviewer rFvy**
> >
> > Thanks to the author for the reply.
> >
> > I still think that comparison on MATH will help better evaluate the framework proposed in this article. GSM8K has been done too well by most open source and closed source LLMs.
> >
> > In addition, only using simple K-8 Level rules to synthesize problems, I think, limits the contribution of this work, because now the community pays more attention to complex reasoning tasks.
> >
> > By the way, Figure2 and Figure3 appear blurry when zoomed in.

---

> > > ### Author Response · Authors · 2024-12-04
> > >
> > > We thank the reviewer for engaging! We hope to clarify the last points below.
> > >
> > > > I still think that comparison on MATH will help better evaluate the framework proposed in this article. GSM8K has been done too well by most open source and closed source LLMs.
> > >
> > > Thank you for your suggestion! We indeed think that a comparison with MATH would be informative, and have thus repeated our analysis from Appendix E / Figure 3 -- where we compared MathCAMPS performance with GSM8K -- to also include MATH. Our results are similar to the result on GSM8k: we see a strong positive correlation (73%) between the two datasets. This indicates that performance on MathCAMPS is still highly predictive of performance on MATH, even if MathCAMPS does not include all the subjects that appear on MATH.
> > >
> > > > In addition, only using simple K-8 Level rules to synthesize problems, I think, limits the contribution of this work, because now the community pays more attention to complex reasoning tasks.
> > >
> > > While the most complex reasoning tasks are necessary to challenge frontier models, we chose to focus on simpler tasks to conduct a more fine grained evaluation of which skills models still lack. For example, NuminaMath-7B, a model that performs well on IMO-level problems (it won the AIMO progress challenge), struggles on the kindergarten task K.CC.C.7, comparing integers between 1 and 10, achieving a 79% on the task. Surprisingly, however, the model achieved 90% accuracy on 4.NF.A.2, the task of comparing fractions with unlike denominators. This discrepancy underscores the importance of evaluating models on simpler tasks, as we do on MathCAMPS: it reveals unexpected weaknesses that may otherwise go unnoticed when focusing exclusively on complex reasoning problems. By identifying such gaps, we can better understand the limitations in models’ foundational skills, which are critical for their ability to generalize and perform reliably across diverse tasks.
> > >
> > > > By the way, Figure2 and Figure3 appear blurry when zoomed in.
> > >
> > > Thank you, we have re-rendered both of them with higher resolution.
> > >
> > > We thank the reviewer again for the engagement and all the suggestions that strengthened our work!

---

### Official Review · Reviewer_FyCa · 2024-11-05

**Soundness:** 3
**Presentation:** 3
**Contribution:** 3
**Rating:** 5
**Confidence:** 4

**Summary:**

The paper proposes a method called MathCAMPS to create mathematical problems grounded in a set of standards set by the Mathematics Common Core from kindergarten through 8th grade. The approach begins by converting each standard into a formal grammar which enables sampling symbolic problems from the grammars with their corresponding answers.

A large language model (LLM) is then used to translate these symbolic problems to math word problems with a cycle-consistency check ensuring that the generated problems are both coherent and faithful. Since the values symbolic problems can be modified the authors also test the robustness of existing models with followup questions (including incremental or counterfactual variations).

The paper includes the release of 9607 problems along with the framework to generate additional ones, and evaluate several of existing language models using this dataset.

**Strengths:**

The paper is clear and well-structured. It introduces a novel approach and provides the framework to generate mathematical reasoning problems using Mathematics Common Core Standards. The flexibility of this approach allows for synthesizing followup question which is used to test the robustness of an extensive list of large language models.

The paper also analyzes checkpoints of Pythia 12B model during pretraining for learning dynamics of mathematical skills.

The extensive evaluation, which includes testing existing models on the dataset and analyzing model performance across different grade levels shows a through and high quality investigation. The paper provides valuable insights into the varying reasoning capabilities of existing LLMs and highlighting significant disparities in their performance.

**Weaknesses:**

While the paper presents extensive evaluations, it often falls short in providing detailed insights or intuitive explanation for the more surprising failures or performance gaps observed in some models. For instance, "surprising failures and gaps in performance" is mentioned twice in the abstract and the intro without much information. A deeper analysis of these areas would offer a richer understanding of the challenges and areas of potential improvement.

In my opinion, Figure 1 should be on page 3 or closer to the reference on page 4 where the method is being discussed. This would improve the flow of the paper and prevent the readers from having to constantly flip back and forth.

A potential limitation of the cycle consistency check is whether it is sufficient on its own. It only verifies the reproduction of the rules, but it does not account for intermediate consistency or potential intermediate issues in the generation. The imperfections of this cycle-consistency check could potentially remove good questions.

The framework only covers 44 reasoning behaviours, with some being very basic and "easy" for current models. This seems relatively low given the complexity of mathematical reasoning. Generating thousands of problems based on just 44 rules seems somewhat repetitive and may limit the diversity of the problems produced.
Higher grades could potentially include and require skills related to lower grades. This could potentially increase the redundancy of generated samples.

The analysis of the follow-up questions appears to be somewhat superficial and lacks depth in exploring the models' performance and reasoning.

**Questions:**

the paper mentions "Prompting the LLM to back-translate the word problem into a symbolic structure and comparing the new answer to the original enables us to eliminate most unfaithful generation errors and maintain high quality." Could you provide some examples of these unfaithful generations? This would help better understand the soundness of the approach.

As mentioned previously, is the cycle consistency sufficient? as it is only checking for the rules and not the intermediate descriptions? Relying solely on the model for back-translation could introduce imperfections, as a strong model might overlook errors in the math word problems and still produce a correct symbolic problem. Or the model might fail to properly adhere to the question's theme, resulting in an awkward or poorly aligned problem.

How do you validate counterfactual follow-ups when modifying variables with fixed limits, such as the number of days in a week or months in a year?

Given that only 44 rules are used to generate thousands of problems, could this limit diversity? How many training examples are needed to effectively learn these 44 rules? I think this experiment is worth studying to assess the quality of the generated examples.

Doesn't a higher grade level inherently include the skills required at lower grade levels?
Also, Shouldn't the average of the levels be weighted instead of uniform over different levels as K level questions are much simpler than 8th grade?

In the counterfactual setting, the second question seems to be merely a variation of the first, with a different value for the variable. How can this truly be considered a "follow-up" if the questions are independent and don't rely on each other? Could these questions not be solved independently?
Did you try just querying the model with the modified question? seems like the old problem and solution serves as a distraction here for the model and not providing much information?

The paper mentions "mathematical dialogue is much less frequent in pre-training data". Not entirely sure if this is true, but most models are instruction tuned and aligned with dialogue data during post-training. Do you think the follow-up setup truly constitutes a mathematical dialogue setup? Since there are only 2 turns.

What is the distribution of 44 skills/standards over classes from k to 8th grade?

"How do mathematical skills develop during pre-training?" This is an interesting question, but how does instruction tuning factor in, given that most current models are fine-tuned/instruction-tuned for this purpose?

Here are some suggestions:
Citations could be removed in Table 1. This is artificially increasing the length.
Figure 2 legends can benefit from a brief description of what each standard means to help readers better interpret the figure.

---

> ### Author Response · Authors · 2024-11-20
>
> We thank the reviewer for their detailed comments and suggestions! We address each point inline below:
>
> > While the paper presents extensive evaluations, it often falls short in providing detailed insights or intuitive explanation for the more surprising failures or performance gaps observed in some models.
>
> We appreciate the reviewer’s suggestion to be more specific about the failures and performance gaps we noticed. We have updated the paper with a new sub-section 4.2, including a few of the standard-wise analyses which we found to be the most surprising. We note that our supplementary material has extensive results that we made easy for users to browse and explore, much more than we had space to discuss in the paper.
>
> > A potential limitation of the cycle consistency check is whether it is sufficient on its own. It only verifies the reproduction of the rules, but it does not account for intermediate consistency or potential intermediate issues in the generation. The imperfections of this cycle-consistency check could potentially remove good questions.
>
> In our manual examination, we found it extremely rare for the cycle consistency to not catch intermediate consistency issues (see Section 3.2: only 2.3% of false positives). To illustrate why, suppose we have a symbolic problem S, generated from our grammar. We then generate a word problem W from S. In cycle consistency, we generate an S' only from W (but not S), and check that both S and S' have the same answer (which is checked symbolically, without the LM). This ensures that, if there was an intermediate issue in generating W (for example, the model invented constants, or inverted a relationship between variables, or generated an ambiguous problem), it's very hard to reconstruct a problem that is equivalent to S. Conversely, we found our false negative rate to be very low (3.3%, also in Section 3.2), so few good questions are filtered out. Of the 7 correct problems that were discarded, we noticed the following distribution: 6 problems had near-correct back translations into a symbolic structure. The error lied in syntax (e.g., a missing closing bracket, or invalid variable name). Only 1 of the 7 good discarded problems had a genuine mathematical error in its back translation. We agree this is informative for readers, and have included a discussion of these error cases in the appendix (section D).
>
> > The framework only covers 44 reasoning behaviours, with some being very basic and "easy" for current models.
>
> We included skills from a range of difficulties to enable true fine-grained measurement of model abilities. So while higher grades do often compose of skills from lower grades, our dataset enables researchers to understand model failures better. For example, if a model is particularly bad at adding fractions, is it because it is bad at addition or fraction comprehension? Knowing what “easy” skills contribute to model failure on more challenging skills allows for a much finer-grained analysis than previous datasets, which only provide aggregate accuracies.
>
> > the paper mentions "Prompting the LLM to back-translate the word problem into a symbolic structure and comparing the new answer to the original enables us to eliminate most unfaithful generation errors and maintain high quality." Could you provide some examples of these unfaithful generations? This would help better understand the soundness of the approach.
>
> Sure! We have added multiple examples of unfaithful problem generations to the appendix, and include two of them here. As you can see, both generated natural language problems are unfaithful to the original structure, and this is caught by the newly generated symbolic structure not having the same final answer.
>
> Original symbolic structure: `[[var f = (5 - 4)]]\n[[var t = (4 - f)]]\n[[question d = ['t']]]\ntheme: Rocket`
> Generated word problem: In the space exploration research center, Diego has 5 rockets. He launched 1 rocket for testing purposes. After this, he orders 4 new rockets, but doesn't count the one that he just launched. How many rockets does Diego have now?
> New symbolic structure: `[[var initial_rockets = 5]]\n[[var launched_rockets = 1]]\n[[var ordered_rockets = 4]]\n[[var total_rockets = (initial_rockets - launched_rockets + ordered_rockets)]]\n[[question result = ['total_rockets']]]`
>
> Original symbolic structure: `[[var v = (79.24 * 37.6)]]\n[[question s = ['v']]]\ntheme: Treasure chest`
> Generated word problem: A pirate finds a treasure chest full of golden coins. Each golden coin weighs 79.24 grams. If the total weight of the coins is 37.6 kilograms, how many golden coins are there in the treasure chest?
> New symbolic structure: `[[var weightInGrams = (37.6 * 1000)]]\n[[var n = (weightInGrams / 79.24)]]\n[[question numCoins = ['n']]]`

---

> > ### Author Response · Authors · 2024-11-20
> >
> > > As mentioned previously, is the cycle consistency sufficient? as it is only checking for the rules and not the intermediate descriptions?
> >
> > In the 245 problems we manually evaluated, all were aligned with the theme we prompted with - maintaining the topic seems to be a very easy task for LLMs. As for a strong model producing correct symbolic problems despite a misaligned natural language question — in the process of back translation, we provide the model with few-shot examples of what back-translation looks like, but crucially do not provide the original symbolic structure. Given this, if a problem is genuinely misaligned, it is very rare that the new symbolic structure has the same final answer as the original one, since the model would have to undo the mistake by chance, without having the information of what the original question looked like.
> >
> > > How do you validate counterfactual follow-ups when modifying variables with fixed limits, such as the number of days in a week or months in a year?
> >
> > We also run cycle-consistency on follow-up questions. To shed light on the reviewer's specific point, we ran an additional analysis where we manually constructed a symbolic structure and main word problem that included a constant (days/year, months/year, etc.). Then, we prompted the model to generate a followup question with a new symbolic structure, where we changed the constant value. In our three tries, the model generated a bad followup once, but cycle-consistency proceeded to catch it. In the other two tries, the model generated valid questions. For example, one structure where we replaced the constant 7 with a 17 resulted in this followup: “Suppose Alex decides to plant 5 little plants every day, but this time he continues planting for 17 days instead of just a week. How many plants does Alex plant in total?”
> >
> > > Given that only 44 rules are used to generate thousands of problems, could this limit diversity?
> >
> > Having 44 CC Standards enforces diversity, because the construction of symbolic structures for each standard is randomized and limited by the constraints related to that standard. In human-curated datasets, this diversity isn’t guaranteed, because the annotator usually doesn’t start with a randomized ground truth. Notably, Table 4 in [1] shows that over 50% of the questions in GSM8K come from only 3 CC standards. Generating an even number of problems per standard ensures that particular standards aren’t over-represented.
> > [1] https://aclanthology.org/2024.findings-emnlp.323.pdf
> >
> > > How many training examples are needed to effectively learn these 44 rules? I think this experiment is worth studying to assess the quality of the generated examples.
> >
> > While we hope MathCAMPS will be helpful for controlled experiments with training and fine-tuning, such experiments would involve design choices that are out of the scope of the current dataset we provide. Our pipeline does not generate reasoning steps in natural language, only the word problem and its final answer (computed from the symbolic problem), which are enough to grade, but not to train a reasoning model. For that, we would have to distill from a stronger model, but then the choice of base model and strong model will affect results in a non-trivial way. Those are rich directions for future work to explore.
> >
> > > Doesn't a higher grade level inherently include the skills required at lower grade levels? Also, Shouldn't the average of the levels be weighted instead of uniform over different levels as K level questions are much simpler than 8th grade?
> >
> > Yes, higher grade levels typically include skills required at lower grades, while adding complexity to them (e.g., by combining them with new skills). The fine-grained separation in our dataset allows users of MathCAMPS to identify specific failure modes of models. If we were just to evaluate models on higher grades, whenever they failed, we would not know whether this is due to a simpler skill that they are missing, or due to the higher-level skills at that grade, which is a differentiation that MathCAMPS can measure. For example, Numina-7B, the current most performant model on the AIMO challenge, only achieves a 79% on K.CC.C.7, the task of comparing integers within 10. However, it achieves a 90% accuracy on 4.NF.A.2, the task of comparing fractions. Our fine-grained analysis shows that models don't necessarily perform as you would expect from human students, i.e. find higher grades harder and earlier grades strictly easier. As for the weighing, we only show one possible analysis, but we think the greatest value in MathCAMPS is the ability to look at individual results, rather than focus on how to aggregate.

---

> > > ### Author Response · Authors · 2024-11-20
> > >
> > > > In the counterfactual setting, the second question seems to be merely a variation of the first, with a different value for the variable. How can this truly be considered a "follow-up" if the questions are independent and don't rely on each other? Could these questions not be solved independently? Did you try just querying the model with the modified question? seems like the old problem and solution serves as a distraction here for the model and not providing much information?
> > >
> > > You are correct that the counterfactual questions only change one variable in the first question. However, the original question often has much more information that is not included in the counterfactual, so answering just based on the counterfactual is not possible. The model has to understand what the new question changes about the original one, which is a different skill than just interpreting a complete question, and we found models to not be robust to this variation even though the mathematical content is generally the same. As a simple concrete example, here is one original question:
> > >
> > > Lara wants to make a necklace. The necklace requires (11/3) feet of yarn. She also wants to add smaller beads which will extend the length of the necklace by (8/30) feet. How many feet of materials will Lara need to make the necklace?
> > >
> > > And here is a counterfactual follow-up question:
> > >
> > > Lara realized that she made a slight miscalculation. The amount of smaller beads she wants to add to the necklace extends its length by (8/28) feet not by (8/30) feet as she initially thought. Given this new information, how many total feet of material will Lara need to make her necklace, before adding the larger beads?
> > >
> > > Note that the follow-up alone is insufficient for answering the question.
> > >
> > > > The paper mentions "mathematical dialogue is much less frequent in pre-training data". Not entirely sure if this is true, but most models are instruction tuned and aligned with dialogue data during post-training. Do you think the follow-up setup truly constitutes a mathematical dialogue setup? Since there are only 2 turns.
> > >
> > > While models are generally trained with dialogue, it's still true that dialogue about mathematical problems is rare. For example, we manually looked at the Anthropic HH dataset, often used in post-training, and could not find examples of dialogues about math problems. While 2-turn dialogues is still arguably a simple setting, our work is, as far as we are aware, the first paper that tries to benchmark this ability for math problem-solving.
> > >
> > > > What is the distribution of 44 skills/standards over classes from k to 8th grade?
> > >
> > > Tables 4-12 show the grade wise CC standards that we included. The number of skills per grade is as follows: K - 4, 1st - 3, 2nd - 6, 3rd - 8, 4th - 9, 5th - 7, 6th - 4, 7th - 3, 8th - 3
> > >
> > > > "How do mathematical skills develop during pre-training?" This is an interesting question, but how does instruction tuning factor in, given that most current models are fine-tuned/instruction-tuned for this purpose?
> > >
> > > One intuition that papers on instruction tuning shared is that post-training doesn't teach new skills to the model, but rather makes it easier for users to surface skills learned in pre-training via natural language instructions. However, few-shot learning has been shown to work even in base models, such as in the original experiments with (non instruction-tuned) GPT-3 [2]. Thus, we probe the development of these skills with few-shot learning, and have found them to develop in Pythia as our analysis shows.
> > >
> > > [2] https://papers.nips.cc/paper/2020/file/1457c0d6bfcb4967418bfb8ac142f64a-Paper.pdf
> > >
> > > > Here are some suggestions: Citations could be removed in Table 1. This is artificially increasing the length. Figure 2 legends can benefit from a brief description of what each standard means to help readers better interpret the figure.
> > >
> > > Thank you for your suggestions! We edited Table 1 and Figure 2 as the reviewer suggested. We appreciate your efforts in helping us make our manuscript clearer.
> > >
> > > We're happy to engage further should the reviewer have more questions.

---

> > > > ### Comment · Reviewer_FyCa · 2024-11-26
> > > >
> > > > thanks for the reply.
> > > >
> > > > > is the cycle consistency sufficient?
> > > >
> > > > My concern was about a strong model ignoring or fixing a problem during backtranslation such that the final answers now match (given an incorrect structure).
> > > >
> > > >
> > > > > issue with constants (days in a week, month, etc)
> > > >
> > > > Which model were these tests conducted on? I’m not sure if three attempts are enough to definitively rule out any potential issues with the method. Additionally, I’m unclear on how the cycle-consistency mechanism identified this error. The concept of constants in cycle-consistency isn’t explicitly constrained—rather, the focus is on ensuring that the variables and logical forms match.
> > > >
> > > > The example you provided doesn’t seem to have any constraints with the constants mentioned. I was mentioning a case where a phrase such as “He plants trees 3 days a week”
> > > >
> > > > > counterfactual
> > > >
> > > > Thank you for providing the example. However, my concern still stands, as there’s no control over how the LLM generates the counterfactual example in response to the modification. It’s possible that the model could construct the question in such a way that it contains all the necessary information to solve it, effectively making the original question irrelevant.
> > > >
> > > > > mathematical dialogue
> > > >
> > > > I don’t think Anthropic HH is a suitable dataset to contain mathematical dialogue (being about helpfulness and harmlessness). Ultrafeedback seems to contain more of such dialogues.
> > > >
> > > > > “How do mathematical skills develop during post-training
> > > >
> > > > Thank you for sharing your insights on this. I believe that post-training techniques have advanced significantly since [2] and Pythia, which could potentially alter the dynamics of skill acquisition for the model. Would it be possible to test this approach on a more recent family of models, such as the PT and IT versions of LLAMA?

---

> > > > > ### Author Response · Authors · 2024-12-04
> > > > >
> > > > > We thank the reviewer for engaging with our response! We apologize for the delay, but hope that our response below can still be taken into account in the post-rebuttal discussion.
> > > > >
> > > > > >> is the cycle consistency sufficient?
> > > > > > My concern was about a strong model ignoring or fixing a problem during backtranslation such that the final answers now match (given an incorrect structure).
> > > > >
> > > > > Thanks for clarifying the question: we think we understand the point of confusion now. The reason why a strong model cannot simply backtranslate while "ignoring or fixing a problem" is that the model is *not given access to the original problem*, only to the generated word problem. Thus, if the word problem is not faithful to the original symbolic problem (for example, it misses some information that was originally present, affecting the answer), the model cannot simply ignore that the information is missing and copy it back: it would have to manufacture the exact same information without any hint as to what it is (or even any hint that anything is missing in the first place, since the unfaithful word problem is most often a sensible problem, just not corresponding to the original structure). Thus, the reason why cycle consistency is reliable in practice is not due to a limitation on the strength of the model doing backtranslation, but rather due to the information bottleneck in the pipeline. While still there is some non-zero probability of false positives, we again refer to our human evaluation in Section 3.2, where we validated that 97.7% of the problems that passed cycle consistency were faithful to the original symbolic problem, and thus to the answer we computed.
> > > > >
> > > > > > Which model were these tests conducted on? I'm not sure if three attempts are enough to definitively rule out any potential issues with the method. Additionally, I'm unclear on how the cycle-consistency mechanism identified this error. The concept of constants in cycle-consistency isn't explicitly constrained—rather, the focus is on ensuring that the variables and logical forms match.
> > > > >
> > > > > We used GPT-4o as our generator also for this test. While the problem the reviewer considered is possible in principle (since we do not explicitly track bounds, e.g., on how many days a week there are), in practice this is extremely unlikely: the vast majority of variables in the symbolic structures end up being translated into values that can vary arbitrarily in the story. To do a more robust test that this is not an issue in practice, we manually inspected all the 456 problems in MathCAMPS that mentioned either “week,” “month,” or “year.” All these questions used times as part of the story (“Sally earned $x after 2 weeks,” “Next month, Josh will climb $y ft,” etc.), not as a unit that would have a bound. We tried our best to find examples with other themes that would showcase this issue, but could not find any, suggesting that in practice this does not happen in our pipeline (thus, any extra step to check for this would not have a measurable effect right now).
> > > > >
> > > > > > Thank you for providing the example. However, my concern still stands, as there's no control over how the LLM generates the counterfactual example in response to the modification. It's possible that the model could construct the question in such a way that it contains all the necessary information to solve it, effectively making the original question irrelevant.
> > > > >
> > > > > We acknowledge that this is indeed possible. Unlike for the units/limits case, we do find instances of this behavior in the dataset, especially in the simpler standards where there isn't much initial information. However, even in that case, we found our counterfactual follow-ups to often reveal robustness issues regardless of the fact that the full information is given. For example, here is an example with DeepSeek 67B:
> > > > > ``` Compare the values of the following numbers by filling in the blank with <, >, or =. 2 _ 9 DeepSeek 67B: 2 < 9
> > > > >
> > > > > Recall the previous problem where we compared the numbers 2 and 9. Now suppose you replace the number 9 with the number 4. What would the comparison look like now? Fill in the blank with <, >, or =. 2 _ 4
> > > > > DeepSeek 67B: 2 > 4 ```

---

> > > > > > ### Author Response · Authors · 2024-12-04
> > > > > >
> > > > > > (continuing)
> > > > > >
> > > > > > ```A starfish has 5 arms. If there were 360 degrees in a circle, how many degrees would be in each section if it were to be equally divided amongst the starfish's arms?
> > > > > > Claude-3 Haiku:
> > > > > > Okay, let's break this down step-by-step:
> > > > > > 1) A starfish has 5 arms.
> > > > > > 2) There are 360 degrees in a circle.
> > > > > > 3) To find the number of degrees in each section, we need to divide the total degrees in a circle (360) by the number of arms (5).
> > > > > > 4) 360 degrees / 5 arms = 72 degrees per section.
> > > > > > ### Answer: 72 degrees
> > > > > >
> > > > > > Continuing from the previous problem, what if the starfish was an unusual species with 17 arms instead of 5? If there were still 360 degrees in a circle, how many degrees would be in each section if it were to be equally divided amongst the starfish's arms?
> > > > > > Claude-3 Haiku:
> > > > > > Okay, let's solve this step-by-step:
> > > > > > 1) The starfish has 17 arms.
> > > > > > 2) There are 360 degrees in a circle.
> > > > > > 3) To find the number of degrees in each section, we need to divide the total degrees in a circle (360) by the number of arms (17).
> > > > > > 4) 360 degrees / 17 arms = 21.176 degrees per section.
> > > > > > Since we can't have a fractional degree, we need to round the answer to the nearest whole number.
> > > > > > ### Answer: 21 degrees
> > > > > > ```
> > > > > > This serves to show that there are still gaps in understanding even simple concepts once you probe models further -- this is especially pronounced in open models, which have been closing the gap in standard datasets (GPT-4o, for instance, gets 100% of the follow-ups correct in this simple comparison standard, showing that it has no such robustness issue).
> > > > > >
> > > > > > > I don't think Anthropic HH is a suitable dataset to contain mathematical dialogue (being about helpfulness and harmlessness). Ultrafeedback seems to contain more of such dialogues.
> > > > > >
> > > > > > We note that Ultrafeedback also does not contain mathematical dialogue, because all of its examples are single-turn answers from a variety of models to *instructions*. Thus, in Ultrafeedback, there are no examples of follow-up questions. That said, Ultrachat does have true dialogue data (Ultrafeedback samples instructions from Ultrachat, but only the initial one, not follow-ups). We again manually inspected the first 500 examples of Ultrachat (https://huggingface.co/datasets/stingning/ultrachat), and found no examples of mathematical turn-2 questions. Although we don't doubt that some of them might exist, we maintain that this kind of data is extremely rare in existing datasets.
> > > > > >
> > > > > > >> “How do mathematical skills develop during post-training
> > > > > > > Thank you for sharing your insights on this. I believe that post-training techniques have advanced significantly since [2] and Pythia, which could potentially alter the dynamics of skill acquisition for the model. Would it be possible to test this approach on a more recent family of models, such as the PT and IT versions of LLAMA?
> > > > > >
> > > > > > Yes, this is a very interesting suggestion! Our main experiments in the paper were already with the chat (post-trained) version of Llama 3. Thus, to assess how post-training might have affected the skill acquisition, we also ran the corresponding base 8B and 70B models, for comparison. We found a large effect of post-training in Llama, in both directions (some skills seemingly get much better, whereas some get worse). One case that gets consistently better at both scales seems to be operations with fractions: in standard 5.NF.A.2 (a 5th grade standard "Solve word problems involving addition and subtraction of fractions [...]"), Llama 3 8B Chat is 27% better than the base model (35 vs 8% accuracy), whereas Llama 3 70B is a surprising 49% better than its base model (62% vs 13%). Thus, this ability only seems to surface after post-training. In contrast, some other abilities seemingly get worse: the ability to do 4-digit addition for example seems to decay (e.g. by 10% on Llama 8B). On average, the base models underperform. We will add a detailed analysis of this comparison on Llama to the appendix, and note it in the discussion about Pythia. Note that this analysis further shows the value of our fine-grained dataset: we can track exactly what skills change (and which don't - the vast majority are within +-2% of the starting performance) during post-training, and further investigate how it affects specific behaviors by looking at the responses.
> > > > > >
> > > > > > Thank you again for the discussion! We do think it strengthened our work in several points, and are thus grateful for the reviewer's engagement.

---

### Meta-Review · Area_Chair_b9WF · 2024-12-23

**Metareview:**

The paper introduces MathCAMPS, a framework for generating fine-grained mathematical problems aligned with K-8 Common Core Standards using symbolic representations and language models, validated through a cycle-consistency method. Strengths include its focus on curriculum-aligned evaluation, a scalable and automated pipeline, fine-grained skill analysis, and novel tasks like mathematical dialogue testing. It also provides insights into skill development during training. However, weaknesses include its limitation to K-8 level problems, reliance on Common Core standards, and lack of comparison with more complex benchmarks like MATH. This paper received mixed scores of 5,6,6,6. I have read the reviews and author responses. I have no problem with the paper’s novelty and the unique contribution of this paper is to ground the model’s performance on math into the human curriculum, so that we can understand the model's math ability in interpretable ways, this stands in contrast with previous benchmarks. My biggest concern is on the simplicity of this dataset as it is only K-8 level, and the paper only compares with GSM8K. However, the performance on GSM8K-level questions is already saturating and this community is moving to more complex problems, therefore, I feel the practical usage of the proposed benchmark will be largely limited. Based on this, I think this paper is really borderline and lean rejection of this work while I will have no objections if the paper gets accepted in the end.

**Additional Comments On Reviewer Discussion:**

Reviewers appreciated the paper's focus on curriculum-aligned evaluation, scalability, and fine-grained analysis but raised concerns about its limitations to K-8 problems, reliance on Common Core standards, and lack of comparisons with more complex benchmarks like MATH. They also questioned the novelty of the approach, the robustness of the cycle-consistency method, and the diversity of generated problems. The authors addressed these by emphasizing the value of fine-grained skill evaluation, clarifying the effectiveness of cycle consistency with detailed examples, discussing extensions to more complex domains, and comparing their framework with related work like GSM-Symbolic and MORE, highlighting its fully automated pipeline and educational grounding. I think the paper’s biggest contribution is on the fine-grained analysis grounded to Common Core standards, yet my main concern is on the simplicity of this dataset as the community is moving to more challenging benchmarks.

---

### Decision · Program_Chairs · 2025-01-22

Reject